# GRAPH REPRESENTATION LEARNING ENHANCED SEMI-SUPERVISED FEATURE SELECTION

## ABSTRACT

Feature selection process is essential in machine learning by discovering the most relevant features to the modeling target. By exploring the potential complex correlations among features of unlabeled data, recently introduced self-supervision-enhanced feature selection greatly reduces the reliance on the labeled samples. However, they are generally based on the autoencoder with sample-wise self-supervision, which can hardly exploit relations among samples. To address this limitation, this paper proposes Graph representation learning enhanced Semi-supervised Feature Selection(G-FS) which performs feature selection based on the discovery and exploitation of the non-Euclidean relations among features and samples by translating unlabeled "plain" tabular data into a bipartite graph. A self-supervised edge prediction task is designed to distill rich information on the graph into low-dimensional embeddings, which remove redundant features and noise. Guided by the condensed graph representation, we propose a batch-attention feature weight generation mechanism that generates more robust weights according to batch-based selection patterns rather than individual samples. The results show that G-FS achieves significant performance edges in 12 datasets compared to ten state-of-the-art baselines, including two recent self-supervised baselines.

## 1 INTRODUCTION

Supervised feature selection is an essential process in machine learning (ML) to identify the most relevant features for the prediction target to build more interpretable and robust models (Liu & Zheng, 2006; Yin et al., 2014). To achieve this goal, supervised feature selections rely on discriminative information encoded in class labels or regression targets to remove irrelevant, redundant and noisy features (Liu & Yu, 2005). However, labels are generally costly and difficult to acquire in many real-world applications. With limited labels, existing feature selection methods, especially deep learning-based solutions, are likely to suffer significant performance deterioration (Venkatesh & Anuradha, 2019).

Although labeled samples are scarce, large volumes of unlabeled data are often readily available (Perez-Riverol et al., 2019). Therefore, increasing attention has been directed to the study of "semi-supervised feature selection" using label signals from labeled data and data distribution or the local structure of both labeled and unlabeled data to evaluate feature relevance (Han et al., 2014). Lee et al. (2021) points out that those structures can help prevent feature selection models from overfitting to noise or selecting redundant features. The difficulty lies in discovering diverse relations from "simple" tabular data that do not have explicit structures, as opposed to the relations found in images and natural languages.

Most research on semi-supervised feature selection focuses on discovering different types of relation between feature and/or samples, for example, graph theory (Lai et al., 2022b), Markov boundary (Xiao et al., 2017), relations between parents and children (Ang et al., 2015) and/or regularization method (Lai et al., 2022a). However, those hand-crafted relation discovery algorithms can hardly express complex relationships among different types of entities (Sheikhpour et al., 2017) and cannot handle large-scale data due to their time-consuming relation discovery process (Chang et al., 2014). Two recent feature selection approaches, A-SFS (Qiu et al., 2022) and SEFS (Lee et al., 2021) relieve the need for custom-design relations with self-supervision. They learn instance-wise inter-feature relations with autoencoder (AE) with two pretext tasks but largely ignore the rich rela-

tions among samples and sample-feature. Those approaches normally focus on learning single types of relation as specified in Fig. 1b and fail to fully exploit other types of relation.

**Motivation:** Figure 1 shows the motivation of this work. Tabular data actually have many types of relations. Except for the feature-label relations, there exist sample-sample relations (similarity vs. dissimilarity), feature-feature relations, and sample-feature (sample-wise feature values) relations. Thus, it is important to find a data structure to represent tabular data and allow us to learn four types of relations. The graph is a general data structure that models explicit or implicit relationships between objects with non-Euclidean space structures (Zhang et al., 2022), state-of-the-art graph neural networks (GNNs) solutions can be used to exploit possible relations from this graph. By distilling complex structural relationships between features and samples into low-dimensional informative embeddings, we can better learn the correlation of features and remove redundancy among features (Li et al., 2022).

**Contributions:** This paper presents a Graph Representation Learning(GRL) enhanced Semi-supervised Feature Selection method (G-FS) that fully utilizes both labeled and unlabeled samples for feature selection. Mapping tabular data into a bipartite graph eliminates the need for hand-made graph structure discovery and fully relies on the GNN to discover latent relations. G-FS learns informative representations for influential features and correlated samples via GRL without any assumption on the data distribution. Compared to the original samples, the learned low-dimensional embeddings eliminate duplicated or highly correlated features and overcome noise interference. Samples reconstructed with embeddings rather than original data are used for feature selection. To alleviate the impacts introduced by a handful of noisy data, this paper proposes a batch-based feature weight generation module that tunes feature weights according to batch-wise feature selection patterns rather than individual samples.

Extensive experiments have been performed on twelve representative real-world datasets to validate our approach. The results show that G-FS discovers relevant features that provide superior prediction performance compared to ten state-of-the-art baselines for both classification and regression tasks. Further analysis of one-shot feature selection shows that G-FS has a much lower demand for labeled data than other methods with better performance.

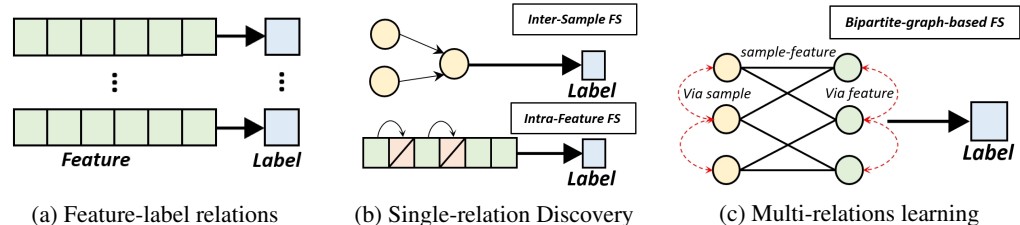

|  (a) Feature-label relations  |  (b) Single-relation Discovery  |  (c) Multi-relations learning  |

Figure 1: Feature selection based on different types of relation discovery, (a) shows the traditional FS which exploits feature-label relations (b) semi-supervised FS with additional single-relation discovery: sample-sample or feature-feature relations (c) G-FS learns all four types of relations: feature-feature, sample-sample, feature-sample and feature-label.

## 2   RELATED WORK

Feature selection is a well-studied problem. Although there are many unsupervised feature selection methods that do not require any labels, such as AEFS (Sheikhpour et al., 2020) and DUFS (Lindenbaum et al., 2021), here we only discuss scenarios where labels are available.

**Supervised feature selection:** The early supervised feature selection for flat data based on information entropy usually assumes that each feature is independent while ignoring their correlation and potential feature structure. They can be generally divided into the wrapper, filter, and embedded methods (Li et al., 2017). Rather than limiting to domain-specific with specific structures, e.g., graph (Jiang et al., 2019), tree (Wang & Ye, 2015), recent trends of supervised feature selection methods are more focused on deep learning, which is considered to have the potential to overcome the 'curse of dimensionality and volume' with its capabilities in encoding different data. Those methods learn to identify feature importance by sparse one-to-one regulation (Li et al., 2016), atten-

tion selection possibility (Gui et al., 2019), pairwise nonlinear transformation (Huang et al., 2020), dual-network model Wojtas & Chen (2020) or with independent Gaussian random variables (Yamada et al., 2020). Reference (Wei et al., 2022) extends feature selection to the control domain with deep reinforcement learning. However, those solutions have a strong reliance on labels and might overfit when only limited labeled samples are available (Kuncheva et al., 2020).

**Semi-supervised feature selection:** Semi-supervised feature selections have been proposed to mitigate label dependence problems by using both unlabeled and labeled data for feature selection. The goal is to use labeled samples to maximize the separability between different categories and to use unlabeled samples to preserve the local data structure with hand-crafted algorithms with specific metrics, e.g., Markov boundary (Xiao et al., 2017) or causality discovery between Parents and Children (Ang et al., 2015). They face significant challenges in learning complex structures with comparable simple metrics, e.g., variance score, laplacian score, fisher score, and constraint scores, and mainly or limited to categorical features (Sheikhpour et al., 2017). Recently, some graph-based methods have been proposed, such as GS3FS (Sheikhpour et al., 2020) and AGLRM (Lai et al., 2022b), which consider the correlation between features, but these methods mainly use predefined structures (for example, graph laplacian matrix) to extend traditional methods rather than graph learning. Furthermore, they cannot handle large-scale data because of the time-consuming computation of the relations (Chang et al., 2014).

**Self-supervised enhanced feature selection:** In recent years, self-supervised learning has gained many research interests as it can create (weak) supervised signals from unlabeled data. Tabnet (Arık & Pfister, 2021) used unsupervised pretraining to predict masked features. It could not use labeled samples. Recent work, A-SFS (Qiu et al., 2022) and SEFS (Lee et al., 2021) train an AE to capture structural information. However, due to the complexity of the latent structure among features and samples, sample-wise AE faces difficulties in capturing relationships among samples. To the best of our knowledge, G-FS is the first deep-learning framework that utilizes GRL with both labeled and unlabeled samples for feature selection.

## 3 G-FS Architecture

In this section, the major notations and designs of G-FS and key modules are illustrated.

### 3.1 Notations

In this paper, value as lowercase character (e.g. $a$); vector as lowercase bold character (e.g. $\mathbf{a}$); matrix as uppercase bold character (e.g. $\mathbf{A}$); set as uppercase italic character (e.g. $A$); Graph as cursive character (e.g. $\mathcal{G}$).

**Partially labeled tabular data:** In feature selection, we have a tabular dataset $\mathbf{D}$ with $n$ samples and $m$ dimensions (features). The $i$-th sample is denoted as $\mathbf{d}_i$, and the $j$-th feature of the $i$-th sample is denoted as $d_{ij}$. Within $\mathbf{D}$, only partial samples are labeled, denoted as a set $\mathbf{D}^l \in \mathbf{D}$. Without loss of generosity, we specify that the first $L$ samples are labeled, corresponding to the label set $Y = \{y_1, y_2, \ldots, y_L\}$.

**Unlabled tabular data vs. bipartite graph:** In order to achieve GRL for tabular data, we translate the tabular data $\mathbf{D}$ into a bipartite graph $\mathcal{G} = (S, V, E)$, with $S$ and $V$ indicate the set of all samples and the set of all features in $\mathbf{D}$, respectively, $S = \{s_1, s_2, \ldots, s_n\}$ and $V = \{v_1, v_2, \ldots, v_m\}$. Note that the symbol $\mathbf{d}_i$ is equal to $\mathbf{s}_i$. $E$ is the edge set where $e_{ij}$ is an edge between $s_i$ and $v_j$: $E = \{(s_i, v_j, e_{ij}) | s_i \in S, v_j \in V, \}$ where the weight of the edge equals the value of the $j$-th feature of sample $i$. Please note, $e_{ij}$ and $d_{ij}$ are interchangeable. We use $d_{ij}$ in the context of data matrix $\mathbf{D}$, and $e_{ij}$ in the context of graph-related tasks.

### 3.2 Architectural design

Similar to SEFS and A-SFS, the feature selection process of G-FS is generally divided into two separate phases: the self-supervision process and the feature selection process.

Figure 2 shows the two-phase architecture of G-FS. In the first phase, the *Bipartite Graph representation learning* module (in the left part of Figure 2) translates the unlabeled tabular data into a

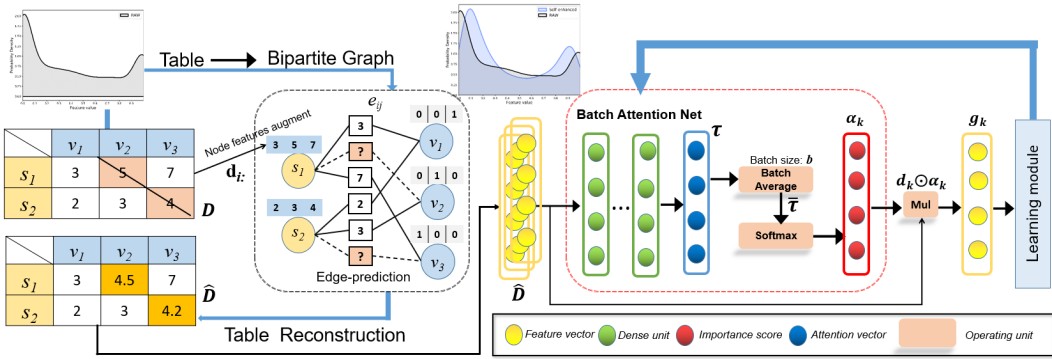

Step 1: Bipartite Graph Representation Learning | Step 2: Batch-attention-based Feature Selection

Figure 2: The two-phase framework of G-FS, the first phase(left), learns the bipartite graph representation with edge prediction, and the second phase(right), learns the batch-based attention weights.

bipartite graph where the samples and features are viewed as two types of nodes in a bipartite graph and the feature values as edges. Then, a pretext task with edge prediction is proposed to learn the low-dimensional embeddings of the bipartite graph. After the graph representation learning process, the masked data with labels $\mathbf{D}^l$ is reprocessed with the trained model to uncover masked features. Those uncovered features contain implicit relations in the data matrix $\mathbf{D}$. The second phase is performed by the *Batch-attention-based Feature Selection* module (in the right part of Figure 2). This module computes the weights for all features with supervisory signals from labels with partially reconstructed labeled data. The following sections illustrate how the two modules are designed.

### 3.3 BIPARTITE GRAPH REPRESENTATION LEARNING

Graph representation learning aims to distill various graph information into dense vector embedding with automatically generated weak supervisory signals. The learned sample representations can, to some extent, explicitly represent both inter-samples relations and sample-wise feature relations.

#### 3.3.1 TABULAR DATA TO BIPARTITE GRAPH

In $\mathcal{G}$, nodes in the sample set $S$ and feature set $V$ do not have features. According to (You et al., 2019b), the node features could help capture structure information (positional information). There are different ways of node feature initialization. Murphy (Murphy et al., 2019) creates node features with one-hot encoding and GRAPE (You et al., 2020) uses constant $\mathbf{1}$ as the sample node features and one-hot encoding for $V$. However, such formulation would make the GNN model hard to differentiate messages from different samples. Thus, we use the i-th row of the tabular data $\mathbf{D}$ as the sample feature value vector $\mathbf{d}_{i:}$ as the sample node $s_i$'s feature, while using the one-hot node features for each $v_j$. The node feature matrix $\mathbf{F}$ is defined as follows:

$$\mathbf{F} = \begin{cases} \mathbf{d}_{i:} & s_i \in S \\ onehot & v_j \in V \end{cases} \tag{1}$$

#### 3.3.2 GRL FOR THE BIPARTITE GRAPH

We apply GRL to bipartite graphs by using the idea of G2SAT (You et al., 2019a) and generally follow the guidelines in GRAPE (You et al., 2020) to define the message-passing process with several enhancements. In the bipartite GRL, we define three types of embeddings: $\mathbf{p}_i$ for sample node $s_i$, $\mathbf{q}_j$ for the feature node $v_j$ and $\mathbf{e}_{ij}$ for edge embedding of $e_{ij}$.

**Messaging Passing:** At the $l$-th GNN layer, the message passing function takes the concatenation of the embedding of the source node $\mathbf{p}_i/\mathbf{q}_j$ and the edge embedding $\mathbf{e}_{ij}$ as input:

$$\mathbf{h}_i^{(l)} \leftarrow Mean \left( \sum_j \sigma \left( \mathbf{W}^{(l)} \cdot Concat \left( \mathbf{q}_j^{(l-1)}, \mathbf{e}_{ij}^{(l-1)} \right) \right) \right) \tag{2}$$

where $\mathbf{W}$ is the trainable weight matrix. $Mean$ denotes the mean operation with a non-linear transformation $\sigma$, Concat is an operation for concatenation. The node embedding $\mathbf{p}_i^{(l)}$ and edge embedding $\mathbf{e}_{ij}^{(l)}$ are updated by:

$$\begin{cases} \mathbf{p}_i^{(l)} = \sigma\left(\mathbf{Q}^{(l)} \cdot Concat(\mathbf{p}_i^{(l-1)}, \mathbf{h}_i^{(l)})\right) \\ \mathbf{e}_{ij}^{(l)} = \sigma\left(\mathbf{P}^{(l)} \cdot Concat\left(\mathbf{e}_{ij}^{(l-1)}, \mathbf{p}_i^{(l)}, \mathbf{q}_j^{(l)}\right)\right) \end{cases} \tag{3}$$

where $\mathbf{P}$ and $\mathbf{Q}$ are the trainable weight matrices. To simplify the illustration, here we only include $\mathbf{p}_i^{(l)}$ in equation (2), $\mathbf{q}_i^{(l)}$ is updated in the same way as $\mathbf{p}_i^{(l)}$. Then the attributes of masked edges are predicted by the corresponding sample embedding $\mathbf{p}_i$ and feature embedding $\mathbf{q}_j$:

$$\hat{D}_{ij} = O_{edge}\left(Concat\left(\mathbf{p}_i^{(l)}, \mathbf{q}_j^{(l)}\right)\right) \tag{4}$$

where $O_{edge}$ is a multi-layer perceptron(MLP).

**Pretext task with edge prediction:** In order to learn GRL, similar to the work (You et al., 2020; 2019a), a self-supervision edge prediction task is proposed to learn latent data structures with $\mathcal{G}$ with certain masked edges for prediction. We randomly mask out a certain percentage of edges, use them as surrogate labels, and use the remaining edges and original sample and feature nodes to predict those surrogate labels. As shown in the left part of Figure 2, the orange blocks represent masked edges that have to be predicted.

Let a binary mask matrix $\mathbf{N} \in \{0, 1\}^{n \times m}$ indicates whether the edge is masked, when $d_{ij}$ is masked, $N_{ij} = 0$. Thus, the informative representation can be learned with $\hat{D}_{ij} = \hat{\mathbf{e}}_{ij}$ by minimizing the difference between $D_{ij}$ and $\hat{D}_{ij}$ for all masked edges with $N_{ij} = 0$. As the masked tabular data might contain both continuous and discrete values, when imputing discrete attributes, we use CE loss for discrete attributes ($\alpha = 1$) and MSE loss for continuous attributes ($\alpha = 0$).

$$\mathcal{L} = \alpha \cdot CE(\mathbf{D}, \hat{\mathbf{D}}) + (1 - \alpha) \cdot MSE(\mathbf{D}, \hat{\mathbf{D}}) \tag{5}$$

where $\hat{\mathbf{D}}$ is the reconstructed data matrix.

### 3.4 BATCH-ATTENTION-BASED FEATURE SELECTION

During the self-supervised GRL process, features from labeled samples are randomly masked and reconstructed. Those reconstructed data are used in the feature selection process. The attention module proposed in AFS (Gui et al., 2019) is used to extract the potential relationship between features and labels. However, in real-world data, the high-noise nature of the data often leads to the performance degradation of the attention mechanism. The sample-wise attention generation might be easily influenced by noise or "poor" samples. Thus, we adopt a batch-based attention generation module inspired by batch-wise attenuation (Liao et al., 2021).

**Attention generation:** For each batch b, an attention vector of different samples can be generated by a two-layer dense network, which compresses $\hat{\mathbf{D}}$ into a vector $\boldsymbol{\tau} = [\tau_1, \tau_2, ...\tau_b]^T \in \mathcal{R}^{b \times m}$, the batched vector $\bar{\boldsymbol{\tau}}$ is averaged by:

$$\bar{\boldsymbol{\tau}} = \frac{1}{|b|} \sum_{k=1}^{|b|} \left(\mathbf{T}_2 \cdot tanh(\mathbf{T}_1 \cdot \boldsymbol{d}_k + \boldsymbol{c}_1) + \boldsymbol{c}_2\right) \tag{6}$$

Where $\boldsymbol{d}_k$ is the sample in a batch, $\mathbf{T}_1$, $\mathbf{T}_2$ are trainable weight matrices, $\boldsymbol{c}_1$, $\boldsymbol{c}_2$ are bias vectors. To generate feature weight, a softmax transformation is used to convert the probabilities of the features selected into an importance score vector with the range $(0, 1)$: $\boldsymbol{\alpha}_k = e^{\bar{\boldsymbol{\tau}}_k} / \sum_{j=1}^m e^{\bar{\tau}_j}$. It allows the importance of different features to be evaluated in the same ranges and enlarges the difference in weights to facilitate feature selection.

**Learning for feature selection:** The sample $\boldsymbol{d}_k$ is multiplied with $\boldsymbol{\alpha}_k$ pairwise $\odot$ to obtain the weighted feature vector $\boldsymbol{g}_k = \boldsymbol{d}_k \odot \boldsymbol{\alpha}_k$, and the weight is adjusted through back-propagation until convergence. Vector $\boldsymbol{g}_k$ are fed into an evaluation network to evaluate the accuracy of feature

importance scores, updated using the loss function below:

$$\arg \min_{\theta_f} \left[ \mathcal{L} \left( \mathcal{F} \left( \boldsymbol{g}_k; \theta_f \right), y_k \right) \right] \tag{7}$$

where $\mathcal{F}\left(\cdot\right)$ is a 3-layer MLP with 64 hidden units with parameters $\theta_f$, $\mathcal{L}\left(\cdot\right)$ is the loss function: CE loss for classification and MSE loss for regression.

## 4 EXPERIMENT

This section evaluates the performance of G-FS with real-world datasets. Source codes, pseudo-codes, detailed settings, datasets descriptions, scalability and robustness analysis, the running time and computational complexity analysis and more extensive experiment results can be found in the Appendix.

### 4.1 EXPERIMENT SETTINGS

**Datasets.** Table 1 shows the basic features of twelve evaluation datasets, including six regression and six classification datasets, taken from UCI ML[1] and OpenML library[2]. As all datasets are fully observed, we randomly select 10% percent of data(unless explicitly specified) as labeled for supervised learning. The number of features with the highest weight (Top-K) is arbitrarily determined as follows: 3% of dataset with more than 200 features and 10% of the data set with fewer than 200 features, with a minimum of 3. The total samples, excluding labels, are used for self-supervised learning.

**Baselines.** G-FS is compared with ten strong feature selection baselines: *ML-based:* LASSO (Tibshirani, 1996), LightGBM (Ke et al., 2017), XGBoost (Chen & Guestrin, 2016) and CCM (Chen et al., 2017). *Semi-supervised:* Semi-JMI and Semi-MIM (Sechidis & Brown, 2018). *DNN-based:* FIR (Wojtas & Chen, 2020) and SANs (Yang et al., 2019). *Self-supervised:* A-SFS (Qiu et al., 2022) and SEFS (Lee et al., 2021). We also compared with unsupervised baselines, FRUFS (Jensen & Shen, 2008) and AEFS (Han et al., 2018), results and descriptions of the baselines can be found in Appendix **B**.

**Settings.** All the baseline algorithms selected for comparison use the default settings proposed in their original paper. For self-supervised methods (A-SFS, SEFS and G-FS), self-supervised iterations and feature selection iterations are 20000. In the GRL phrase, we use a 3-layer GNN with 64 hidden units and RELU activation, the edge mask ratio is 30%, and the optimizer is Adam with a learning rate of 0.001. The batch attention net with 64 hidden units is optimized by Adam, with a learning rate of 0.002, and the batch size is 128.

**Evaluation Metrics.** For real-world data, we do not have ground-truth feature relevance. Like in the previous literature, we use the prediction performance of TopK features selected by different feature selection methods to assess the quality of selected features. LightGBM is used for evaluation to avoid possible preferences for DNN-based solutions. It also has a more stable performance than MLP. The experiment is repeated 10 times using random seeds ranging from 0~9 and the data is split into training and testing sets with a ratio of 7:3. The Micro-F1 score is used (in %) for classification tasks, while the mean absolute error (MAE) is used for regression tasks.

### 4.2 EXPERIMENT RESULTS

To verify the performance of G-FS, G-FS is compared with other feature selection methods on 12 different datasets (refer to Table 1). Semi-JMI and Semi-MIM can only work for classification, and SEFS fails to generate weights for three datasets.

#### 4.2.1 PERFORMANCE COMPARISONS

**Regression tasks:** With limited labels (10% of the original number), it becomes difficult for existing methods to identify the most relevant features. Their performance tends to be unstable, especially

---
[1]https://archive.ics.uci.edu/
[2]https://www.openml.org/

Table 1: Performance comparison. The upper part for regression tasks(MAE↓, lower the better), and the lower part for classification tasks(Micro-F1↑, higher the better). − means no result

| Algor. | Fea./Top-K | LASSO | LGB | XGB | CCM | FIR | SANs | SemiJMI | SemiMIM | A-SFS | SEFS | G-FS |
|---|---|---|---|---|---|---|---|---|---|---|---|---|
| MBGM↓ | 359/11 | 6.50 | 5.98 | 5.62 | − | 8.99 | 7.27 | − | − | 8.34 | − | **5.46** |
| Pdgfr↓ | 320/10 | .168 | .166 | .167 | .156 | .147 | .153 | − | − | .144 | − | **.131** |
| Tecator↓ | 124/12 | 2.20 | 1.13 | **0.89** | 1.66 | 1.40 | − | − | − | 1.40 | 0.99 | 1.03 |
| CPU↓ | 21/3 | 5.37 | 5.34 | 2.47 | 4.16 | 6.37 | 4.34 | − | − | 3.04 | 4.01 | **2.37** |
| Protein↓ | 9/3 | 3.95 | 3.71 | 3.77 | 3.81 | 4.22 | 4.21 | − | − | 3.99 | 3.75 | **3.68** |
| Concrete↓ | 8/3 | 5.33 | 5.15 | 5.56 | 7.02 | 8.24 | 5.74 | − | − | 5.38 | 5.54 | **4.96** |
| CIFAR10↑ | 3072/92 | 28.09 | 40.89 | 41.15 | 40.97 | 41.45 | 40.26 | 32.87 | 31.80 | 35.92 | 39.93 | **41.92** |
| Micro.↑ | 1300/39 | 27.00 | 23.67 | 23.67 | 18.67 | 22.10 | 25.33 | 28.18 | 31.33 | 30.54 | − | **36.50** |
| MNIST↑ | 784/24 | 50.22 | 55.63 | 51.65 | 42.46 | 36.15 | 30.23 | 50.88 | 51.27 | 57.45 | 55.56 | **58.67** |
| Isolet↑ | 618/18 | 66.39 | 71.80 | 68.48 | 57.52 | 64.07 | 57.61 | 55.77 | 56.97 | 64.79 | 70.76 | **73.12** |
| USPS↑ | 256/8 | 75.16 | 80.58 | 81.02 | 80.61 | 77.90 | 79.64 | 74.34 | 73.38 | 81.59 | 81.22 | **83.71** |
| Optdigits↑ | 64/6 | 73.55 | 77.59 | 69.67 | 59.36 | 58.05 | 63.69 | 72.46 | 71.43 | 78.22 | 77.66 | **79.46** |

DNN-based SANs and FIR. In comparison, the self-supervision enhanced solutions, A-SFS, SEFS and G-FS, generally achieve good performance, as they can learn the latent feature structures in the tabular data to avoid overfitting or noise impacts. However, the AE-based solutions used in SEFS and A-SFS might not be able to capture the rich structures existing in the samples.

**Classification tasks:** Decision tree-based method such as XGB is susceptible to noise values. The increase in leaves makes them sensitive to overfitting. The complex DNN structures make it difficult for FIR and SANs to be trained with a limited number of labeled samples. Thus, their performances are even worse than ML-based solutions. Semi-supervision methods, which are difficult to handle complex high-dimensional data and are limited to a small number of labels, have poor performance. Self-supervision methods achieve even higher performance edges than other solutions as they have unlabeled data for structure learning. G-FS achieves further performance gains on both datasets than SEFS and A-SFS thanks to its rich relations discovery capabilities of graphs.

**Why does G-FS work?** We compare the raw data $\mathbf{D}$ and the reconstructed data $\hat{\mathbf{D}}$(A-SFS and G-FS), 25% features of Optdigits are selected. Those data are projected into a two-dimensional space for visualization by T-SNE (see Figure 5). Compared to $\mathbf{D}$, $\hat{\mathbf{D}}$ is observed to have a more compact embedding and represent more distinct category boundaries. This can be attributed to the ability of self-supervised mechanisms to learn more structured representations from tabular data than from raw values. Compared with AE (Fig. 5b), the embedding generated by bipartite graph (Fig. 5c) can better aggregate samples of the same type together and achieve a higher silhouette score. It might explain the superior performance of G-FS over AFS with AE-based self-supervision.

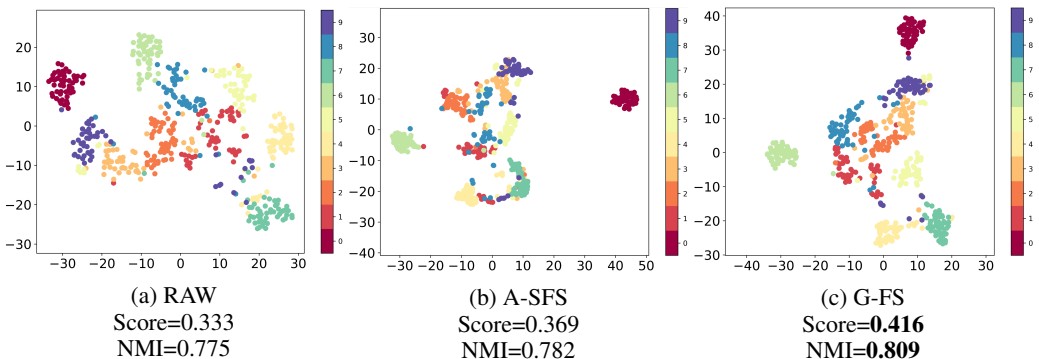

(a) RAW
Score=0.333
NMI=0.775

(b) A-SFS
Score=0.369
NMI=0.782

(c) G-FS
Score=**0.416**
NMI=**0.809**

Figure 3: T-SNE distribution of Top-16 raw/reconstructed features of Optdigits, NMI means normalized mutual information score; Scores means silhouette score, two indicators the higher the better.

### 4.2.2 IMPACTS FROM LABELED AND UNLABELED SAMPLES

In this part, we consider the impact of labeled and unlabeled samples from the self-supervised methods SEFS, A-SFS, and G-FS. As seen in Figures 4a and 4b, when the number of samples is small, G-FS already has a good performance advantage over A-SFS and SEFS. It clearly shows the effec-

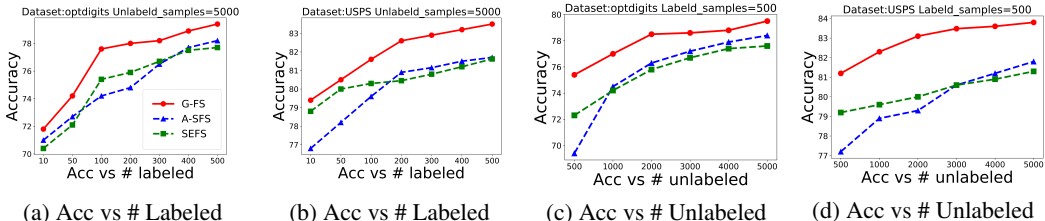

(a) Acc vs # Labeled    (b) Acc vs # Labeled    (c) Acc vs # Unlabeled    (d) Acc vs # Unlabeled

Figure 4: Accuracy on Optdigits (Top-6) and USPS (Top-8). (a) and (b) varying numbers of labeled samples; (c) and (d) varying numbers of unlabeled samples.

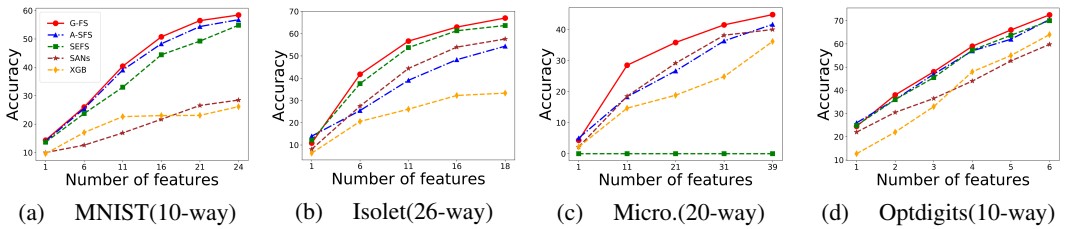

(a)  MNIST(10-way)    (b)  Isolet(26-way)    (c)  Micro.(20-way)    (d)  Optdigits(10-way)

Figure 5: One-shot feature selection on 4 classification datasets with different Top-K features. G-FS demonstrates a consistent superiority in performance regardless of the number of K features.

tiveness of G-FS in the use of labeled samples. When the number of labeled samples increases, G-FS still maintains a lead in most cases, which shows that G-FS can already find relevant features with a limited number of samples. Similarly, increasing the number of unlabeled samples, as shown in Figure 4c and 4d, generally helps. For the range 500∼5000, G-FS maintains a consistent advantage over SEFS and A-SFS. SEFS and A-SFS, due to their AE solutions, have limitations in inter-sample learning. G-FS, in comparison, performs better with more unlabeled samples.

### 4.2.3 ONE-SHOT FEATURE SELECTION

The ability to learn object categories from a few examples is critical in ML. We study the generalization abilities of SOTA methods with one sample per class. The task demands that those methods have strong generalizability. The performance of different baselines is shown in Figure 5 with the different number of top-ranked features selected for classification. When K increases, G-FS keeps a constant increase in performance. At the same time, major supervised solutions suffer poor performance, as one sample per class can hardly provide enough signals to give the appropriate weights. A-SFS, SEFS, and G-FS show good performance for one-shot learning, while G-FS generally outperforms the other two in most feature ranges. This experiment shows that G-FS can effectively weight high-dimensional features even in the one-shot learning setting with limited labels.

### 4.3 RESULTS ON SYNTHETIC DATA

In this part, we create a synthetic dataset using scikit-learn library by generating point clusters of vertices of a 10-dimensional hypercube (following a normal distribution with std=1) according to 10 relevant features and assign the same number of clusters to each class (supervised target Y). We then added 240 redundant features, which were linear combinations of the relevant features. Thus, there are 250 related features. Then 250 features with random numbers are added and makes the total feature 500. To make the FS process more challenging, we randomly mask 30% features to 0.

Table 2: Performance comparison(Micro-F1↑) on the synthetic dataset, Count/ 15 means the number of relevant features found in TOP-15 features.

| Top-K | LASSO | LGB | XGB | SANs | FIR | SemiJMI | SemiMIM | A-SFS | G-FS |
|---|---|---|---|---|---|---|---|---|---|
| 5 | 17.53±5.98 | 20.60±3.10 | 21.20±3.19 | 11.73±2.48 | 23.60±5.39 | 13.18±2.02 | 19.33±5.64 | 22.40±2.25 | **23.96**±3.10 |
| 10 | 22.26±4.38 | 27.13±5.49 | 28.66±2.78 | 14.40±2.09 | 23.20±2.64 | 18.26±4.83 | 28.73±5.77 | **33.66**±2.10 | 33.06±2.14 |
| 15 | 22.53±5.71 | 31.53±3.78 | 33.40±4.22 | 21.86±6.76 | 20.93±4.70 | 26.73±5.62 | 34.33±3.53 | 39.13±2.09 | **39.50**±1.66 |
| Count / 15 | 9.1 | 10.3 | 10.9 | 6.5 | 6.9 | 7.8 | 9.6 | 13.7 | **15.0** |

We compared the baseline algorithms with different Top-K and the results are presented in Table 2. G-FS achieved the highest accuracy at the Top-5 and Top-15 levels and the second highest accuracy at the Top-10 level. It can find all 15 relevant features (or redundant features) in Top-15, while XGB only 10.9. A-SFS achieves the second accuracy, and SEFS cannot generate results on the synthetic dataset. It clearly shows the effectiveness of G-FS.

### 4.4 G-FS STRUCTURE ANALYSIS

A set of experiments is designed to check the effectiveness of our designs.

**Ablation Studies.** We evaluate the mechanisms of self-supervision and batch attention generation with two variants: 1) GFS$^{-g}$: removes the self-supervision and only uses batch attention networks for weight generation. 2) G-FS$^{-b}$: keeps the self-supervision while removing the batch attention networks. Figure 6 clearly shows that both designs are important for performance improvements. G-FS$^{-g}$ can identify the most influential features, but weak in identifying combined influences with multiple features. G-FS$^{-b}$ input only one sample at a time and easily disturbed by noise.

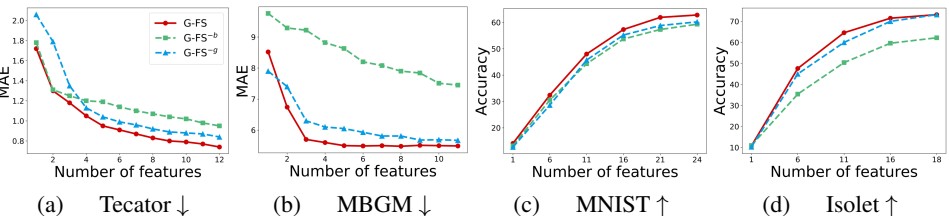

|     |     |     |     |     |     |     |     |
|-----|-----|-----|-----|-----|-----|-----|-----|
| (a) | Tecator ↓ | (b) | MBGM ↓ | (c) | MNIST ↑ | (d) | Isolet ↑ |

Figure 6: Ablation study on both regression and classification datasets

**Feature selection under different GNN architecture.** This part verifies that GNN structures have significant impacts on relation learning. We compare G-FS with the EGCN and a heterophily-based model IGRM Zhong et al. (2023). EGCN extends GCN Kipf & Welling (2016) with added edge information. The results under different GNN layers are put in Appendix D.

Table 3: Performance comparison with different GNN structure

| Architecture | Optdigits↑ | USPS↑ | MBGM↓ | Tecator↓ |
|---|---|---|---|---|
| G-FS(G2SAT) | **79.08**±2.57 | **83.10**±1.46 | **5.45**±0.11 | **1.03**±0.17 |
| G-FS(IGRM) | 75.51±2.02 | 80.71±0.98 | 6.12±0.44 | 1.08±0.21 |
| G-FS(EGCN) | 72.50±3.56 | 81.14±2.07 | 5.90±0.35 | 1.16±0.14 |

Results in Table 3 shows that the EGCN and IGRM has significant performance degradation. The reason contributing to this is that EGCN mixes the embeddings of the ego sample and the neighbor sample, which cause a loss of important information due to the different semantic information contained in the samples and features. IGRM containing too much bias toward homogeneity and heterophily will decrease the performance of FS. In contrast, G2SAT concatenates the sample ego and neighbor embeddings rather than mixing them and can better keep different semantic information.

## 5 CONCLUSION

This paper proposes G-FS, a novel feature selection framework that can utilize the large volume of often readily available unlabeled data with graph representation learning to discover latent structures among features and samples. We translate plain tabular data into a bipartite graph with node feature augmentation and use one self-supervised edge prediction task to encode bipartite graphs into low-dimensional embeddings. This process can largely remove noise and express complex correlations among features with low-dimensional embeddings. Then, we use the batch attention mechanism to generate feature weights with reconstructed data. In experiments on twelve real-world datasets, we validate that our model discovers features that provide superior prediction performance on both classification and regression datasets. Further experiments prove that G-FS can effectively select features in one-shot feature selection settings. One of our future works is to extend the GRL to a general tabular representation learning method to support various downstream tasks.

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

APPENDIX

## A: Dataset descriptions

Table 1 shows the basic features of twelve evaluation datasets, including six regression and six classification datasets, taken from UCI ML[3] and OpenML library[4].

Table 1: Dataset descriptions

| Datasets | Features | TopK | Classes | Labeled | Total |
|---|---|---|---|---|---|
| MBGM | 359 | 3%(11) | – | 421 | 4209 |
| Pdgfr | 320 | 3%(10) | – | 8 | 79 |
| Tecator | 124 | 10%(12) | – | 24 | 240 |
| CPU | 21 | 3 | – | 819 | 8192 |
| Protein | 9 | 3 | – | 4573 | 45730 |
| Concrete | 8 | 3 | – | 103 | 1030 |
| CIFAR10 | 3072 | %3(92) | 10 | 1000 | 10000 |
| Micro. | 1300 | %3(39) | 20 | 20 | 200 |
| MNIST | 784 | %3(24) | 10 | 200 | 2000 |
| Isolet | 618 | %3(18) | 26 | 260 | 2600 |
| USPS | 256 | %3(8) | 10 | 500 | 5000 |
| Pixel. | 240 | %3(8) | 10 | 200 | 2000 |
| GAS | 128 | %3(5) | 6 | 600 | 6000 |
| Optdigits | 64 | %10(6) | 10 | 500 | 5000 |

Here is a detailed description of all datasets:

- MBGM, short for Mercedes Benz Greener Manufacturing, thisis a highly anonymized dataset released by Mercedes-Benz, which aims to predict the time a car spends on the test bench during production. This dataset contains 4209 rows and 360 columns, where the last column specifies the target variable, i.e., the time a car spends on the test bench. Each row in this dataset represents a single car produced by Mercedes-Benz, and each column describes a feature that characterizes the car. This dataset is a challenge due to its high dimensionality, large number of categorical features, and highly unbalanced target variable. In addition, it also has a high class imbalance ratio, which makes the problem more challenging for machine learning models.

- Pdgfr, this is one of 41 drug design datasets. The datasets with 1143 features are formed using Adriana.Code software. The molecules and outputs are taken from the original studies.

- Tecator, this dataset is a spectroscopic dataset that contains measurements of the absorbance spectra of 240 samples of pork meat. These data are recorded on a Tecator Infratec Food and Feed Analyzer working in the wavelength range 850 - 1050 nm by the Near Infrared Transmission (NIT) principle. Each sample contains finely chopped pure meat with different moisture, fat, and protein contents. For each meat sample, the data consists of a 100-channel spectrum of absorbances and the contents of moisture (water), fat, and protein. The absorbance is -log10 of the transmittance measured by the spectrometer. The three contents, measured in percent, are determined by analytic chemistry.

- CPU, short for Computer Activity database, is a collection of computer systems activity measurements. The data was collected from a Sun Sparcstation 20/712 with 128 Mbytes of memory running in a multi-user university department. Users would typically be doing a large variety of tasks ranging from accessing the internet, editing files or running very cpu-bound programs. The data was collected continuously on two separate occasions. On both occasions, system activity was gathered every 5 seconds. The final dataset is taken from both occasions with equal numbers of observations coming from each collection epoch.

---

[3]https://archive.ics.uci.edu/
[4]https://www.openml.org/

- Protein, short for Physicochemical Properties of Protein Tertiary, Structure. This dataset is taken from CASP 5-9. There are 45730 decoys and size varying from 0 to 21 armstrong.

- Concrete, this is a regression dataset with high noise value. Concrete is the most important material in civil engineering. The concrete compressive strength is a highly nonlinear function of age and ingredients. These ingredients include cement, blast furnace slag, fly ash, water, superplasticizer, coarse aggregate, and fine aggregate.

- CIFAR10, this is a popular benchmark dataset in computer vision. It consists of 3072 features in 10 different classes. CIFAR-10 dataset is widely used to evaluate and compare the performance of different machine learning and deep learning algorithms. It provides a challenging task of classifying small, low-resolution images with high intra-class and inter-class variations. The 10 classes in the CIFAR-10 dataset are: airplane, automobile, bird, cat, deer, dog, frog, horse, ship, and truck.

- Micro, short for Micro_mass, this dataset aims to explore machine learning approaches for the identification of microorganisms from mass-spectrometry data. A reference panel of 20 Gram positive and negative bacterial species covering 9 genera among which several species are known to be hard to discriminate by mass spectrometry (MALDI-TOF). Each species was represented by 11 to 60 mass spectra obtained from 7 to 20 bacterial strains, constituting altogether a dataset of 571 spectra obtained from 213 strains. The spectra were obtained according to the standard culture-based workflow used in clinical routine in which the microorganism was first grown on an agar plate for 24 to 48 hours before a portion of the colony was picked, spotted on a MALDI slide and a mass spectrum was acquired.

- MNIST, this is a widely used dataset of handwritten digits, with each example being a 28x28 grayscale image of a single digit ranging from 0 to 9.

- Isolet, short for Isolated Letter Speech Recognition, this dataset was generated as follows: 150 subjects spoke the name of each letter of the alphabet twice. All attributes are continuous, real-valued attributes scaled into the range -1.0 to 1.0. The features include spectral coefficients; contour features, sonorant features, pre-sonorant features, and post-sonorant features. The exact order of appearance of the features is not known.

- USPS, this dataset contains normalized handwritten digits, automatically scanned from envelopes by the U.S. Postal Service. The original scanned digits are binary and of different sizes and orientations; the images here have been inclined and size normalized, resulting in 16 x 16 grayscale images.

- Optdigits, this is an optical recognition of handwritten digits dataset, 32x32 bitmaps are divided into non-overlapping blocks of 4x4 and the number of on pixels are counted in each block. This generates an input matrix of 8x8 where each element is an integer in the range 0~16. This reduces dimensionality and gives invariance to small distortions.

- Pixel, short for Mfeat_pixel, this dataset describes features of handwritten numerals (0~9) extracted from a collection of Dutch utility maps. The maps were scanned in 8-bit grey value at density of 400dpi, scanned, sharpened, and thresholded. Corresponding patterns in different datasets correspond to the same original character. 200 instances per class (for a total of 2,000 instances) have been digitized in binary images.

- GAS, short for gas drift, this dataset was gathered within January 2007 to February 2011 (36 months) in a gas delivery platform facility situated at the ChemoSignals Laboratory in the BioCircuits Institute, University of California San Diego. This archive contains 13910 measurements from 16 chemical sensors utilized in simulations for drift compensation in a discrimination task of 6 gases at various levels of concentrations.

As all datasets are fully observed, we randomly select 10% percent of data as labeled for feature selection. Top-K in table 1 means the K features with the highest weights. The total samples with labels removed are used for self-supervised learning. From the perspective of data transmission, the workflow of G-FS is shown in Figure 1.

## B: Baselines

The performance of G-FS is compared with nine state-of-the-art feature selection baselines:

**ML-based:**

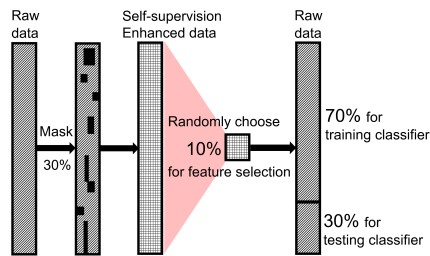

Figure 1: The work flow of G-FS

- LASSO (Tibshirani, 1996), is a model that adds the constraint term of L1 norm after the cost function of the linear regression model.
- LGB (Ke et al., 2017), short for LightGBM, a highly efficient gradient boosting decision tree.
- XGB (Chen & Guestrin, 2016), short for XGBoost, a scalable tree gradient boosting decision tree.
- CCM (Chen et al., 2017), a feature selection method that employs conditional covariance minimization.

**DNN-based:**

- FIR (Wojtas & Chen, 2020), a DNN-based feature selection method with a dual-net architecture.
- SANs (Yang et al., 2019) a model for feature importance evaluation based on self-attention network architecture.

**Semi-supervised:**

- Semi-JMI (Sechidis & Brown, 2018), a semi-supervised based feature selection method which can use "soft" prior knowledge to decide the optimal features subset.
- Semi-MIM (Sechidis & Brown, 2018), a semi-supervised based feature selection method that adopts mutual information to measure each feature's relevancy to the class label, which does not consider redundancy and complementariness among features.

**Self-supervised:**

- A-SFS (Qiu et al., 2022), an attention-based self-supervision-enhanced feature selection method with an autoencoder to uncover the structural among features.
- SEFS (Lee et al., 2021), a self-supervision-enhanced feature selection method with correlated gates and an autoencoder via two pretext tasks.

**Unsupervised:**

- FRUFS (Jensen & Shen, 2008), is an unsupervised feature selection method that conducts feature selection based on fuzzy similarity relations
- AEFS (Han et al., 2018), which selects the most important features by obtaining linear and nonlinear information among the features.

## C: Experiment Settings

**Settings.** All the baseline algorithms selected for comparison use their default settings proposed in their original paper. For self-supervised methods(A-SFS, SEFS, and G-FS), self-supervised iterations are 20000. In the GRL phrase, we use a 3-layer GNN with 64 hidden units and RELU activation, the edge mask ratio is 30%, and the optimizer is Adam with a learning rate of 0.001. The batch attention net with 64 hidden units is optimized by Adam, with a learning rate of 0.002, and the batch size is 128.

Throughout the experiments, for the self-supervised step, training G-FS takes approximately a few minutes to a few hours on a single GPU machine[5]. For the feature selection step, training G-FS only takes about few minutes on a CPU machine[6].

**Evaluation Metrics.** For real-world data, we do not have ground-truth feature relevance. Like previous literature, we use the prediction performance of Top-K features selected by different feature selection methods to assess the quality of selected features. The LightGBM is used for evaluation to avoid possible preferences for DNN-based solutions. It also has a more stable performance than MLP. The experiment is repeated 10 times using random seeds ranging from 0∼9, and the data is split into training and testing sets with a ratio of 7:3. The Micro-F1 score is used(in %) for the classification tasks while the mean absolute error(MAE) for regression tasks.

**Pseudo code:** The pseudo code of self-supervised GRL algorithm and batch attention FS algorithm please refer to Algorithm 1 and Algorithm 2.

---

**Algorithm 1** Self-supervised graph representation learning for G-FS

**Input**: bipartite graph $\mathcal{G} = (S, V, E)$, dataset $\mathbf{D}$, iteration K
**Output**: $\hat{\mathbf{D}}$

1: $t \leftarrow 0$
2: $\mathbf{p}^{(0)} \leftarrow \mathbf{d}_{i:}$
3: $\mathbf{q}^{(0)} \leftarrow onehot$
4: $\alpha \leftarrow \mathbf{1}$ **if** Classification **else 0**
5: **while** $k < K$ **do**
6: $\quad \mathbf{h_i}^{(l)} \leftarrow GNN_1(\mathbf{q_j}^{(l-1)}, \mathbf{e_{ij}}^{l-1})$
7: $\quad \mathbf{p_i}^{(l)}/\mathbf{q_i}^{(l)} \leftarrow GNN_2(\mathbf{p_i}^{(l-1)}/\mathbf{q_i}^{(l-1)}, \mathbf{h_i}^{(l)})$
8: $\quad \mathbf{e_{ij}}^{(l)} \leftarrow GNN_3(\mathbf{e_{ij}}^{(l-1)}, \mathbf{p_i}^{(l)}, \mathbf{q_j}^{(l)})$
9: $\quad D_{pred} \leftarrow MLP(\mathbf{p_i}^{(l)}, \mathbf{q_j}^{(l)})$
10: $\quad \mathcal{L} = \alpha \cdot CE(D_{pred}, D) + (1 - \alpha) \cdot MSE(D_{pred}, D)$
11: $\quad k \leftarrow k + 1$
12: **end while**

---

**Algorithm 2** Batch attention feature selection

**Input**: dataset $\mathbf{D}$, batchsize $b$,iteration K
**Output**: $\alpha$

1: $k \leftarrow 0$
2: $i \leftarrow 0$
3: $\beta \leftarrow \mathbf{1}$ **if** Classification **else 0**
4: **while** $k < K$ **do**
5: $\quad$ **while** $i < b$ **do**
6: $\quad\quad \mathbf{d}_k = [d_1, d_2, ..., d_b] \leftarrow Sampling(\mathbf{D})$
7: $\quad\quad i \leftarrow i + 1$
8: $\quad$ **end while**
9: $\quad \boldsymbol{\tau}_k \leftarrow DenseLayer_2(DenseLayer_1(\mathbf{d}_k))$
10: $\quad \boldsymbol{\alpha}_k \leftarrow softmax(\sum_{i=1}^{|b|} \tau_k)$
11: $\quad \boldsymbol{g}_k \leftarrow \boldsymbol{d}_k \odot \boldsymbol{\alpha}_k$
12: $\quad \boldsymbol{y}_{pred} \leftarrow MLP(\boldsymbol{g}_k)$
13: $\quad \mathcal{L} = \boldsymbol{\beta} \cdot CE(\boldsymbol{y}_{pred}, \boldsymbol{y}) + (1 - \boldsymbol{\beta}) \cdot MSE(\boldsymbol{y}_{pred}, \boldsymbol{y})$
14: $\quad k \leftarrow k + 1$
15: **end while**

---

[5]GPU – NVIDIA TITAN RTX
[6]CPU – Intel Core i7-7700

Table 2: Overall performance with Catboost regressor/classifier. The upper part is the regression tasks(MAE↓, lower the better), and the lower part is the classification tasks(Micro-F1↑, higher the better).

| Algor. | Fea./Top-K | LASSO | LGB | XGB | CCM | FIR | SANs | Semi-JMI | Semi-MIM | A-SFS | SEFS | G-FS |
|---|---|---|---|---|---|---|---|---|---|---|---|---|
| MBGM↓ | 359/11 | 6.56±1.02 | 6.25±0.49 | 5.53±0.26 | – | 8.94±0.05 | 7.88±1.15 | – | – | 5.69±0.74 | – | **5.49**±0.15 |
| Pdgfr↓ | 320/10 | .142±.027 | .143±.026 | .157±.021 | .138±.017 | .161±.011 | .148±.015 | – | – | .148±.016 | – | **.130**±.020 |
| Tecator↓ | 124/12 | 1.87±0.23 | 0.99±0.08 | 0.76±0.08 | 1.06±0.39 | 2.55±0.11 | 1.26±0.33 | – | – | 0.93±0.24 | 0.83±0.04 | **0.74**±0.08 |
| CPU_act↓ | 21/3 | 5.34±0.09 | 5.27±2.52 | 2.43±0.14 | 3.52±1.57 | 8.18±0.07 | 4.24±1.81 | – | – | 2.77±0.79 | 3.02±0.45 | **2.31**±0.02 |
| Protein↓ | 9/3 | 3.87±0.01 | **3.63**±0.01 | 3.70±0.01 | 3.84±0.27 | 4.35±0.29 | 4.01±0.22 | – | – | 3.86±0.10 | 3.76±0.09 | 3.65±0.09 |
| Concrete↓ | 8/3 | 4.51±0.63 | **4.67**±0.43 | 4.94±0.64 | 7.18±2.14 | 9.93±0.23 | 8.16±2.86 | – | – | 7.44±2.14 | 7.13±2.54 | 4.82±0.46 |
| CIFAR10↑ | 3072/92 | 30.15±0.99 | 43.18±1.23 | 43.99±0.96 | 43.36±0.67 | 44.03±0.95 | 42.70±2.44 | 35.26±2.58 | 34.01±2.41 | 40.92±0.69 | 41.46±1.57 | **45.40**±0.99 |
| Micro.↑ | 1300/39 | 37.33±8.06 | 34.66±6.82 | 36.16±5.82 | 40.56±4.83 | 42.66±6.07 | 44.01±8.03 | 34.67±5.60 | 35.67±1.39 | 43.66±5.06 | – | **50.56**±7.32 |
| MNIST↑ | 784/24 | 52.57±3.72 | 62.98±2.10 | 46.26±5.33 | 46.70±4.35 | 36.49±7.65 | 31.91±6.30 | 55.79±0.65 | 55.47±2.12 | 62.43±2.42 | 60.61±4.20 | **63.10**±2.35 |
| Isolet↑ | 618/18 | 70.62±1.68 | 74.20±4.24 | 72.98±4.09 | 70.21±4.67 | 64.30±4.69 | 67.25±5.19 | 54.61±4.03 | 55.50±3.55 | 63.82±5.93 | 66.03±6.28 | 72.20±1.81 |
| USPS↑ | 256/8 | 78.20±2.55 | 80.58±4.16 | 81.82±2.60 | 81.13±3.12 | 81.60±2.63 | 80.52±3.55 | 77.55±0.67 | 74.64±0.90 | 83.85±1.35 | 83.09±1.97 | **85.35**±1.73 |
| Pixel↑ | 240/8 | 62.56±6.02 | 68.95±3.88 | 68.50±2.79 | 61.73±5.08 | 64.93±4.94 | 54.98±3.75 | 62.63±4.85 | 55.66±3.86 | 71.20±2.50 | 71.06±4.29 | **77.66**±3.06 |
| GAS↑ | 128/5 | 91.34±0.61 | 90.91±2.57 | 94.65±1.83 | 94.37±2.31 | 94.22±2.67 | 94.56±1.98 | 89.89±3.76 | 86.91±3.43 | 95.26±1.07 | 96.89±0.84 | **97.29**±0.50 |
| Optdigits↑ | 64/6 | 74.73±2.87 | 79.51±2.37 | 66.38±5.32 | 77.70±4.67 | 58.12±7.07 | 60.32±10.20 | 73.90±0.82 | 73.44±2.32 | 81.36±1.53 | 79.92±2.57 | **81.58**±2.40 |

Table 3: Overall performance with LightGBM regressor/classifier. The upper part is the regression tasks(MAE↓, lower the better), and the lower part is the classification tasks(Micro-F1↑, higher the better).

| Algor. | Fea./Top-K | FRUFS | AEFS | LASSO | LGB | XGB | CCM | FIR | SANs | Semi-JMI | Semi-MIM | A-SFS | SEFS | G-FS |
|---|---|---|---|---|---|---|---|---|---|---|---|---|---|---|
| MBGM↓ | 359/11 | 8.89±0.12 | – | 6.50±0.99 | 5.98±0.06 | 5.62±0.15 | – | 8.99±0.25 | 7.27±1.19 | – | – | 8.34±0.75 | – | **5.46**±0.11 |
| Pdgfr↓ | 320/10 | .166±.019 | – | .168±.017 | .166±.019 | .167±.011 | .156±.018 | .147±.005 | .153±.018 | – | – | .144±.012 | – | **.131**±.015 |
| Tecator↓ | 124/12 | 1.13±0.13 | – | 2.20±0.21 | 1.13±0.13 | **0.89**±0.15 | 1.66±0.33 | 2.94±0.08 | 1.40±0.27 | – | – | 1.40±0.37 | 0.99±0.13 | 1.03±0.17 |
| CPU↓ | 21/3 | 8.12±0.19 | – | 5.37±0.10 | 5.34±2.52 | 2.47±0.17 | 4.16±1.26 | 6.37±0.20 | 4.34±1.87 | – | – | 3.04±1.03 | 4.01±0.71 | **2.37**±0.14 |
| Protein↓ | 9/3 | 4.51±0.01 | – | 3.95±0.01 | 3.71±0.08 | 3.77±0.06 | 3.81±0.09 | 4.22±0.35 | 4.21±0.33 | – | – | 3.99±0.11 | 3.75±0.09 | **3.68**±0.12 |
| Concrete↓ | 8/3 | 10.28±0.39 | – | 5.33±0.39 | 5.15±0.38 | 5.56±0.79 | 7.02±2.05 | 8.24±2.72 | 5.74±1.07 | – | – | 5.38±0.62 | 5.54±0.63 | **4.96**±0.29 |
| CIFAR10↑ | 3072/92 | 21.27±0.84 | 24.53±0.17 | 28.09±0.92 | 40.89±1.24 | 41.15±0.64 | 40.97±0.64 | 41.45±0.34 | 40.26±1.41 | 32.87±3.63 | 31.80±2.74 | 35.92±1.88 | 39.93±1.62 | **41.92**±0.83 |
| Micro.↑ | 1300/39 | 23.66±5.21 | 25.46±1.57 | 27.00±5.95 | 23.67±5.20 | 23.67±5.20 | 18.67±3.85 | 22.10±7.37 | 25.33±6.94 | 28.18±3.41 | 31.33±3.79 | 30.54±9.31 | – | **36.50**±7.20 |
| MNIST↑ | 784/24 | 13.94±1.21 | 42.49±4.30 | 50.22±3.39 | 55.63±3.32 | 51.65±4.41 | 42.46±2.41 | 36.15±5.60 | 30.23±5.72 | 50.88±0.58 | 51.27±1.42 | 57.45±2.92 | 55.56±1.87 | **58.67**±2.45 |
| Isolet↑ | 618/18 | 33.46±2.44 | 49.93±3.74 | 66.39±4.74 | 71.80±4.49 | 68.48±3.50 | 57.52±8.07 | 64.07±4.19 | 57.61±6.19 | 55.77±4.17 | 56.97±5.02 | 64.79±6.92 | 70.76±1.49 | **73.12**±2.72 |
| USPS↑ | 256/8 | 42.16±0.75 | 60.70±2.62 | 75.16±1.65 | 80.58±4.16 | 81.02±2.60 | 80.61±3.30 | 83.77±0.56 | 79.64±4.03 | 74.34±2.65 | 73.38±1.39 | 81.59±1.46 | 81.22±2.59 | **83.71**±1.95 |
| Pixel↑ | 240/8 | 28.00±1.15 | 48.73±4.60 | 56.78±3.33 | 68.11±3.48 | 67.26±3.15 | 61.26±4.45 | 63.28±5.39 | 53.56±3.83 | 62.06±5.49 | 55.93±4.62 | 69.80±3.02 | 70.67±4.29 | **74.67**±2.38 |
| GAS↑ | 128/5 | 79.95±0.80 | 96.50±0.71 | 90.91±0.59 | 90.62±2.61 | 94.53±1.12 | 94.11±2.59 | 93.05±2.71 | 94.40±2.09 | 89.67±3.80 | 85.26±3.89 | 94.92±1.28 | 96.71±0.89 | **97.24**±0.48 |
| Optdigits↑ | 64/6 | 40.88±1.38 | 55.99±8.98 | 73.55±3.52 | 77.59±2.34 | 69.67±2.37 | 59.36±7.10 | 58.05±11.01 | 63.69±6.54 | 72.46±0.56 | 71.53±2.12 | 78.22±1.76 | 77.66±2.06 | **79.46**±2.09 |

# D: More experiments

**Performance comparisons on different classifier** In this section, we replace LightGBM with Catboost. Both the Catboost regressor and classifier are set by default. Table 2 shows the performance comparisons for both types of datasets. Table 3 shows the LightGBM results with standard deviation.

**Regression tasks:** With a limited number of labels(original 10%), it becomes difficult for existing feature selection methods to identify the most relevant features. Their performance tends to be unstable, especially DNN-based SANs and FIR. In comparison, the self-supervision enhanced solutions, A-SFS, SEFS, and G-FS, generally achieve good performance as they can learn the latent feature structures in the tabular data to avoid over-fitting or noise impacts. However, the autoencoder-based solutions used in SEFS and A-SFS might not be able to capture rich structures existing in samples.

**Classification tasks:** Decision tree-based methods such as XGB and RF are susceptible to noise values. The increment of leaves makes them sensitive to overfitting. The complex DNN structures make FIR and SANs hard to be trained with a limited number of labeled samples. Thus, their performances are even worse than ML-based solutions. Semi-supervision methods, which are difficult to handle complex high-dimensional data and limited to a small number of labels, have poor performance. In those datasets, the self-supervision methods achieve even higher performance edges than solutions with no supervision as they have more unlabeled data for self-supervision. G-FS achieves further performance gains on both datasets than SEFS and A-SFS thanks to its rich relations discovery capabilities of graphs.

## Ablation studies

In this section, we perform the ablation studies on more datasets. Figure 2 clearly shows that both designs are important for performance improvements. We can see that G-FS$^{-g}$ with only batch-attention feature selection can effectively identify the most influential features, but it lacks the capability to identify the combined influences with multiple features. G-FS$^{-b}$ can only input one sample at a time, it is easily disturbed by noise.

## Impacts from labeled and unlabeled samples

We compare the impacts from both labeled and unlabeled samples for self-supervised methods SEFS, A-SFS, and G-FS on MNIST. As seen from Fig. 3a, when labeled samples increase, G-FS still maintains a lead in most cases. It shows that G-FS can already find relevant features with

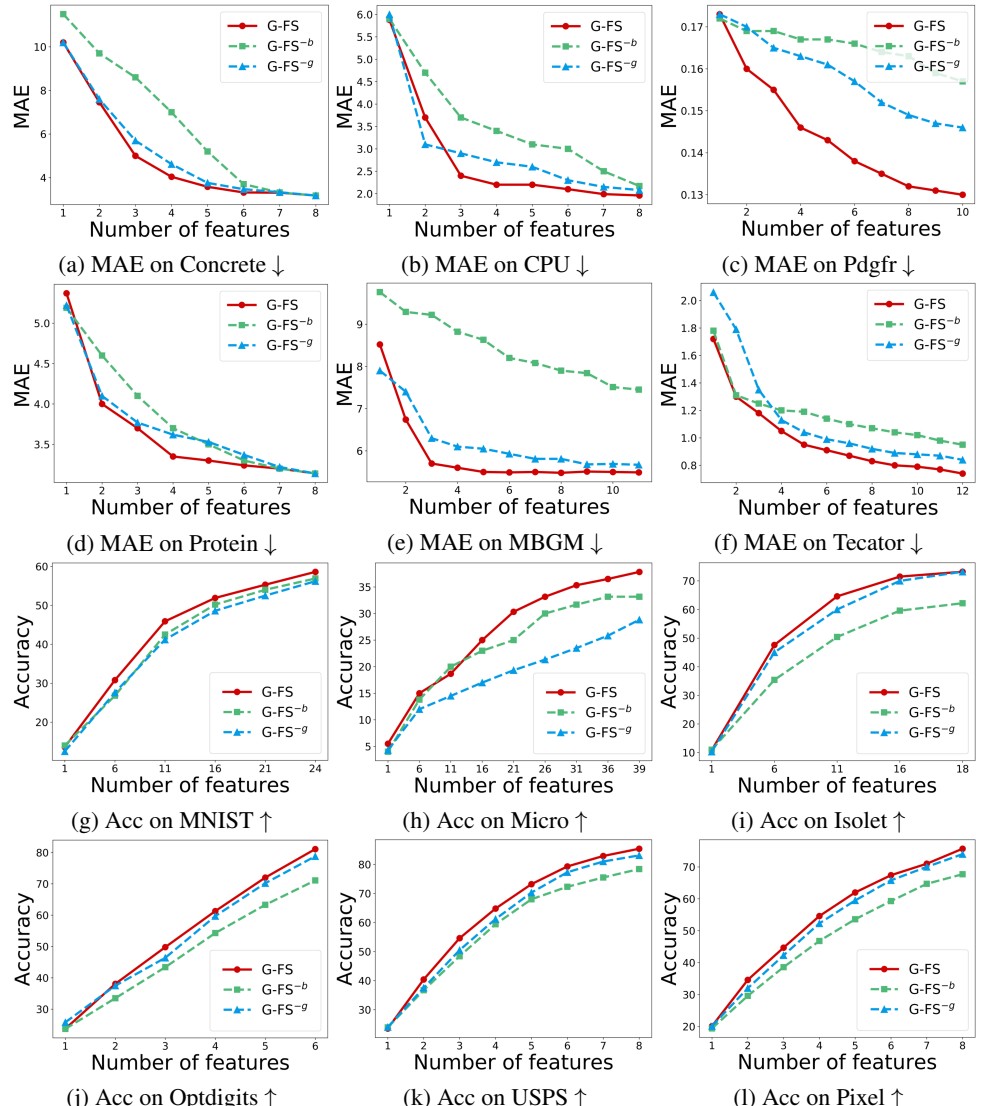

Figure 2: Ablation study on all twelve datasets, (a)∼(f) are regression datasets, and (g)∼(l) are classification datasets.

a limited number of samples. On the other hand, increasing the number of unlabeled samples, as shown in Fig. 3b, generally helps. For the range 800∼2000, G-FS maintains a consistent advantage over SEFS and A-SFS. SEFS and A-SFS, due to their AE solutions, have limitations in inter-sample learning. G-FS, in comparison, performs better with more unlabeled samples.

**A more stable version of G-FS: for classification**

Through more experiments, we found that replacing the 3-layer MLP in the batch attention module with a 4-layer MLP can achieve faster convergence and more stable results (please refer to Table 4). However, this variant does not work well in the few shot scenario.

**Feature selection under different GNN architecture**

In this section, we compare G-FS with different GNN architectures, G2SAT and EGCN, to see whether GNN structures significantly impact feature selection. Table 5 shows the results on twelve regression and classification datasets.

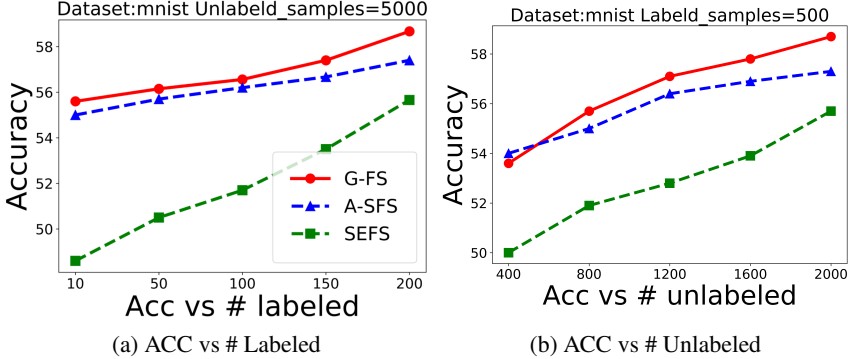

(a) ACC vs # Labeled         (b) ACC vs # Unlabeled

Figure 3: Results on MNIST with TOP-24 features. Accuracy with (a) varying numbers of labeled samples given 2000 unlabeled samples and (b) varying numbers of unlabeled samples given 200 labeled samples.

Table 4: GFS_mlp3 v.s. GFS_mlp4

| Architecture | G-FS(3layer-MLP) | G-FS(4layer-MLP) |
|---|---|---|
| Micro | **36.50**±7.20 | 33.67±8.02 |
| MNIST | 58.67±2.45 | **58.92**±1.63 |
| USPS | 83.71±1.95 | **84.17**±1.44 |
| Pixel | 74.67±2.38 | **78.40**±3.31 |
| Isolet | 73.12±2.72 | **73.49**±2.65 |
| Optdigits | **79.46**±2.09 | 79.07±2.22 |

Results in Table 5 show that EGCN variant suffers around 2%∼50% performance degradation, compared to G2SAT. The reason contributing to this performance difference is that EGCN mixes the ego-sample embeddings with neighbor sample embeddings. However, as samples and features contain different semantic information, this mixture might lose important information. While G2SAT concatenates the sample ego and neighbor embeddings rather than mixing them and can better keep different semantic information.

**Distribution comparison** In this part, we use the USPS dataset and plot the distribution of raw data and reconstructed data by G-FS, the results shown in Figure 4 demonstrate that, after the self-supervision enhancement step, the reconstructed data maintain a distribution that is largely similar to the original data, with two peaks that are more pronounced.

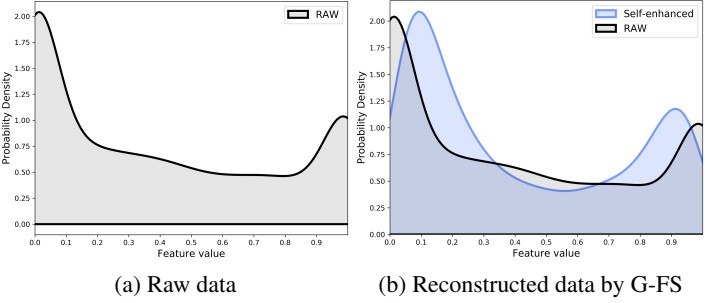

(a) Raw data         (b) Reconstructed data by G-FS

Figure 4: Distribution comparison: Raw v.s. Self-supervision enhancement.

**T-SNE visualization**

In this part, we provide more T-SNE visualization results on different datasets. We compare the raw data $\mathbf{D}$ and the reconstructed data $\hat{\mathbf{D}}$(A-SFS and G-FS), For each dataset, 25% features are selected. And those data are projected into a two-dimensional space for visualization using the T-SNE algorithm (refer to Figure 5). Then we use the k-means algorithm to perform the clustering

Table 5: Performance comparison of G-FS and G-FS(EGCN)

| Architecture | G-FS | G-FS(EGCN) |
|---|---|---|
| Micro | **36.50**±7.20 | 35.50±13.85 |
| MNIST | **58.67**±2.45 | 51.83±2.67 |
| USPS | **83.71**±1.95 | 81.14±1.50 |
| Pixel | 74.67±2.38 | **76.38**±2.21 |
| Isolet | **73.12**±2.12 | 69.21±2.56 |
| Optdigits | **79.46**±2.09 | 72.50±3.56 |
| MBGM | **5.45**±0.14 | 5.90±0.35 |
| Tecator | **1.03**±0.09 | 1.16±0.14 |
| Pdgfr | **.131**±.005 | .169±.021 |
| Concrete | **4.99**±0.63 | 8.96±2.41 |
| Protein | **3.65**±0.09 | 4.05±0.15 |
| CPU_act | **2.43**±0.81 | 3.42±1.10 |

Table 6: Performance comparison under different GNN layers.

| GNN_layer | Optdigits($\uparrow$) | USPS($\uparrow$) | Concrete($\downarrow$) | Tecator($\downarrow$) |
|---|---|---|---|---|
| 1 | 73.46±3.45 | 80.89±2.74 | 6.18±0.52 | 1.16±0.19 |
| 2 | 76.82±3.01 | 81.65±1.82 | 5.31±0.40 | 1.08±0.11 |
| 3 | **79.08**±2.57 | **83.10**±1.46 | **4.96**±0.29 | **1.03**±0.17 |
| 4 | 75.67±2.72 | 81.20±2.61 | 5.46±1.26 | 1.06±0.14 |
| 5 | 69.90±4.13 | 79.60±2.47 | 7.34±2.53 | 1.11±0.13 |

task. The NMI (short for normalized mutual info score) and Scores(short for silhouette score) of all datasets are reported in the caption. Compared to $\mathbf{D}$, $\hat{\mathbf{D}}$ is observed to have a more compact embedding and represent more distinct category boundaries. This can be attributed to the ability of self-supervised mechanisms to learn more structured representations from tabular data than tabular data with raw values. Compared to AE (Figs. 5b, 5e, and 5i), the embedding generated by bipartite graph (Figs. 5c, 5f, and 5f) can better aggregate samples of the same type together and achieve a higher silhouette score. It might explain the superior performance of G-FS over AFS with AE-based self-supervision.

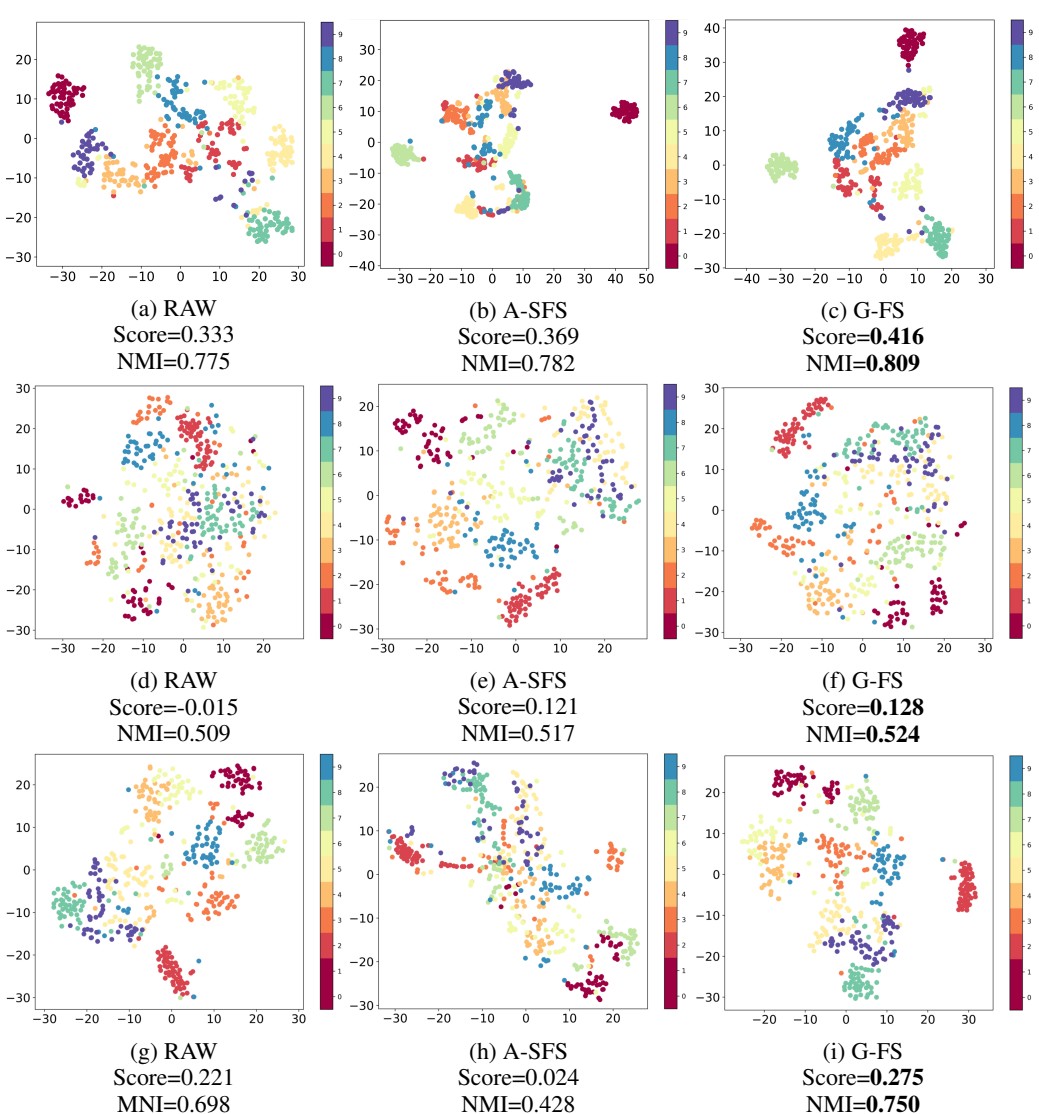

Figure 5: T-SNE distribution of original/reconstructed features of three different datasets, (a)-(c) is optdigits, (d)-(f) is MNIST, (g)-(i) is USPS.

# E: Scalability and robustness analysis

### Feature selection under missing data

In real-world data, missing data is common due to factors such as sensor failures. Therefore, it is necessary to overcome the disturbance caused by the missing data.

To verify whether G-FS can effectively identify reliant features under data with missing, we simulated data missing scenarios in real-world data by masking in a different level(10%,30% and 50%) and set the value with 0. Results with different mask ratios are shown in Table.7. XGBoost is known to be tolerant of missing data, but it generally has poor feature selection performance on classification tasks. Also, DNN-based feature selection solutions are hard to train with missing data, so the performance is not good. Self-supervised solutions, A-SFS, SEFS, and G-FS, keep comparable good performance under data missing due to their masking strategy for structure learning. At missing levels 10% and 30%, G-FS achieves a lower MAE and the highest accuracy in most datasets, outperforming all other benchmark algorithms.

Table 7: Feature selection performance under different data missing ratio(Ra.). The upper part is the regression datasets(MAE↓), the lower part is the classification datasets(Micro-F1↑).

| Algor. | Ra. | XGB | FIR | SANs | A-SFS | SEFS | G-FS |
|---|---|---|---|---|---|---|---|
| Concrete ↓ | 0.1 | **6.20** | 10.14 | 8.75 | 9.62 | 6.92 | 6.58 |
| | 0.3 | **7.35** | 10.28 | 8.46 | 9.47 | 7.56 | 9.05 |
| | 0.5 | **7.69** | 10.20 | 9.48 | 9.97 | 9.14 | 9.55 |
| Tecator ↓ | 0.1 | 1.09 | 2.94 | 1.19 | 1.51 | **1.06** | 1.09 |
| | 0.3 | **1.11** | 2.94 | 1.46 | 1.68 | 1.31 | 1.19 |
| | 0.5 | 1.33 | 2.94 | 1.45 | 1.56 | 1.66 | **1.28** |
| Pdgfr ↓ | 0.1 | .163 | **.143** | .152 | .150 | — | .155 |
| | 0.3 | .163 | **.147** | .159 | .159 | — | .157 |
| | 0.5 | .164 | **.149** | .161 | .158 | — | .154 |
| Protein ↓ | 0.1 | 3.84 | 4.29 | 4.10 | 3.97 | 3.81 | **3.78** |
| | 0.3 | **3.85** | 4.22 | 4.05 | 4.01 | 3.88 | **3.85** |
| | 0.5 | 3.98 | 4.29 | 4.08 | 3.99 | **3.91** | 3.96 |
| CPU ↓ | 0.1 | 3.02 | 8.32 | 4.73 | **3.01** | 4.36 | 4.29 |
| | 0.3 | 3.49 | 8.38 | 4.81 | **3.28** | 4.36 | 4.54 |
| | 0.5 | 4.77 | 8.35 | 5.48 | **3.47** | 4.69 | 4.64 |
| MBGM ↓ | 0.1 | **5.51** | 9.06 | 7.54 | 8.26 | — | 5.54 |
| | 0.3 | **5.72** | 9.04 | 7.84 | 8.47 | — | 5.88 |
| | 0.5 | **5.81** | 9.04 | 7.80 | 8.40 | — | 6.25 |
| MNIST ↑ | 0.1 | 42.30 | 34.85 | 33.08 | 56.93 | 53.53 | **58.35** |
| | 0.3 | 38.48 | 31.70 | 32.66 | 56.63 | 50.26 | **57.26** |
| | 0.5 | 34.53 | 34.05 | 28.71 | 55.40 | 48.63 | **56.53** |
| USPS ↑ | 0.1 | 78.29 | 79.30 | 78.99 | 80.86 | 80.94 | **82.97** |
| | 0.3 | 75.37 | 76.48 | 78.60 | 78.79 | 79.70 | **80.88** |
| | 0.5 | 76.11 | 79.21 | 76.37 | 77.70 | 78.21 | **80.21** |
| Optdigits ↑ | 0.1 | 65.20 | 54.34 | 62.58 | 76.03 | 76.69 | **76.80** |
| | 0.3 | 57.31 | 52.39 | 56.38 | 74.47 | 72.30 | **74.64** |
| | 0.5 | 47.38 | 49.80 | 54.56 | **73.27** | 69.18 | 71.02 |
| Isolet ↑ | 0.1 | 68.16 | 63.79 | 55.65 | 59.61 | 65.82 | **71.14** |
| | 0.3 | 65.85 | 61.46 | 53.61 | 58.67 | 63.02 | **70.57** |
| | 0.5 | 64.82 | 59.92 | 52.92 | 57.89 | 62.54 | **69.19** |
| Pixel ↑ | 0.1 | 65.05 | 60.33 | 59.48 | 67.88 | 66.60 | **73.01** |
| | 0.3 | 61.15 | 63.13 | 52.81 | 67.03 | 65.30 | **69.75** |
| | 0.5 | 55.65 | 61.15 | 51.93 | 63.88 | 65.66 | **67.88** |
| Micro ↑ | 0.1 | 23.67 | 16.66 | 26.50 | 24.50 | — | **38.00** |
| | 0.3 | 23.67 | 20.66 | 26.16 | 22.16 | — | **36.83** |
| | 0.5 | 23.67 | 19.16 | 24.83 | 20.66 | — | **36.33** |

### Discussion on the scalability of G-FS

With the rapid growth of the amount of data, it is important whether an algorithm can be extended to a large scale. Dividing data into batches for training is a common method. Batch operations can be

performed in a self-supervised process if large amounts of unlabeled data are available. Because the node/edge information in the bipartite graph is different in each iteration, it needs a larger iteration step to ensure convergence. Be careful not to make the batch size too small; otherwise, it will cause performance degradation(discussed in the Impacts from labeled and unlabeled samples section). We changed the number of iterations from 20000 to 40000, the number of unlabeled samples from 5000 to 9298. The accuracy of USPS(TOP-8) with 500 unlabeled data is $81.49\pm2.01$, while USPS(TOP-8) with batch size 500 is **82.57**$\pm$**1.57**.

**Runtime comparison**

In this part, we will compare the running time (in seconds) on all algorithms. The two steps, self-supervised enhancement and feature selection, are reported in Table 8 and Table 9. As Table 9 shows, graph self-supervision takes around four times longer than AE self-supervision, with an approximate duration of 6 hours for MNIST. Graphs are much more intricate than simple samples, yet the feature selection process only needs to be done once and can significantly reduce the model's complexity, storage, and computing requirements. Despite the extra computation needed, the significant performance improvements make it worth it.

Table 8: Running times for feature selection on MNIST dataset.

| Time(s) | AEFS | FRUFS | LASSO | RFE | LGB | XGB | CCM |
|---|---|---|---|---|---|---|---|
| MNIST | 18.44 | 12.86 | 219.11 | 258.98 | 0.45 | 0.34 | 225.56 |

| Time(s) | FIR | SANs | Semi-JMI | Semi-MIM | A-SFS | SEFS | GFS(Ours) |
|---|---|---|---|---|---|---|---|
| MNIST | 65.38 | 192.61 | 10.72 | 11.24 | 106.58 | 125.67 | 105.73 |

Table 9: Running times for self-supervised enhancement on MNIST dataset.

| Time(s) | A-SFS | SEFS | GFS(Ours) |
|---|---|---|---|
| MNIST | 6367.48 | 6550.64 | 22018.81 |

# F: Impacts from inconsistent distributions

The previous experiments assumed that the self-supervised data has the same distribution as the supervised data. What would happen if G-FS was self-supervised on unlabeled samples from a distribution different from the supervised samples?

Table 10: Self-supervision on unlabeled data(class 0~4), a different distribution from the labeled samples

| TOP K | 1 | 2 | 3 | 4 | 5 | 6 |
|---|---|---|---|---|---|---|
| G-FS | 23.76 | 39.24 | 46.84 | 59.55 | 72.82 | 79.46 |
| G-FS$^{-g}$ | 21.07 | 30.88 | 44.85 | 55.40 | 64.50 | 70.93 |
| G-FS(0-4) | 25.30 | 44.85 | 55.10 | 63.93 | 76.16 | 80.85 |
| G-FS(5-9) | 20.19 | 28.08 | 35.72 | 48.56 | 60.28 | 66.99 |
| G-FS(0-9) | 22.75 | 36.47 | 45.41 | 56.24 | 68.22 | 73.92 |

We use Optdigits with only class 0~4 samples for self-supervision. This model is then used to reconstruct the data with class 0~9 for the normal feature selection process. Table 10 shows the results of the F1 score for different classes with different Top-K features. For a better comparison, we also put G-FS and G-FS-g results. As we can see, the results for class 0~4 are much better than G-FS, which is self-supervised with balanced data with class 0~9. It clearly shows that class 0~4 benefits from self-supervision. However, class 5~9 suffers from this self-supervision process. They have even worse results than G-FS-g without self-supervision. The overall performance of the unbalanced self-supervision G-FS(0-9) is 1%~3% worse than G-FS with balanced data. It shows the importance of maintaining the consistent distribution for the self-supervision and supervised stage.

