# OpenReview forum: "Graph Representation Learning enhanced Semi-supervised Feature Selection"
_ICLR.cc/2024/Conference — Submitted to ICLR 2024_

### Official Review · Reviewer_s9w1 · 2023-10-17

**Soundness:** 2 fair
**Presentation:** 3 good
**Contribution:** 3 good
**Rating:** 5
**Confidence:** 4

**Summary:**

This article focuses on semi-supervised feature selection. The authors argued that the existing methods can hardly exploit the  relations among samples, so they proposed a  graph representation learning approach named G-FS. G-FS could capture the non-Euclidean relations among features and samples with a a bipartite graph strategy. G-FS achieves significant performance edges in 12 datasets compared with baselines.

**Strengths:**

1. The paper is well organized.
2. Feature selection is important for many areas, the motivation is sound.
3. The experiments are comprehensive.

**Weaknesses:**

1. The proposed model will introduce more parameters by using GNN. The efficiency should be investigated.
2. Compared with XGB, the improvement brought by G-FS is not that significant. It should be discussed.
3. The visualization of tSNE has certain degrees of randomness, it is not sure the result in Figure 3 is convincing since the advantage is small.

**Questions:**

1. The proposed model will introduce more parameters by using GNN. The efficiency should be investigated.
2. Compared with XGB, the improvement brought by G-FS is not that significant. It should be discussed.
3. The visualization of tSNE has certain degrees of randomness, it is not sure the result in Figure 3 is convincing since the advantage is small.

---

> ### Author Response · Authors · 2023-11-13
> **Response to Reviewer s9w1**
>
> Thank you for all your suggestions, we will answer your questions one by one regarding these weaknesses/problems.
>
> **Weakness 1:** The proposed model will introduce more parameters by using GNN. The efficiency should be investigated.
>
> **Answer to Weakness 1:** Thanks for your suggestions. Using GNN to study the relations indeed introduced certain complexity. We also provided the complexity analysis of the GNN's parameters in the original version of Appendix E. Scalability and robustness analysis. Table 7 and Table 8 shows the feature selection time and self-supervision time. Compared to autoencoder-based self-supervision, GFS demands about 4,5 times of times for training in MNIST which is already a large dataset. Considering the benefits of GNN-based self-supervision and this self-supervision only needs to be performed once, we think this overhead is tolerable.
>
> **Weakness 2:** Compared with XGB, the improvement brought by G-FS is not that significant. It should be discussed.
>
> **Answer to Weakness 2:** Thanks for your suggestions.
> For typical supervised classification and regression tasks, G-FS achieves performance improvements. For classification tasks, we achieved the best performance on different types of datasets with an average of 20.32% than all baselines and led the second strong baseline 1.13%\~19.50%. We guess your concerns are about the results of the regression tasks. The regression task is relatively simple and most FS baselines achieve similar results. However, G-FS still achieves good results, especially for MBGM and Pdgfr (two difficult datasets with dim>300), G-FS leads 2.93%\~9.92%. For CPU, Protein, and Concrete, they are comparably simple. G-FS still leads 0.82%\~4.22% in MAE than the second best XGBoost. XGBoost is a very strong baseline in the regression task [1]. Many recent FS do not compare the regression datasets with XGBoost. If XGBoost is excluded, our performance gain can be further improved.
>
> [1] Shwartz-Ziv R, Armon A. Tabular data: Deep learning is not all you need[J]. Information Fusion, 2022, 81: 84-90.
>
> **Weakness 3:** The visualization of tSNE has certain degrees of randomness, it is not sure the result in Figure 3 is convincing since the advantage is small.
>
> **Answer to Weakness 3:** Thank you for your concern. T-SNE is used to demonstrate the effectiveness of G-FS. The advantage of G-FS(about 73%) is much better than the one with raw data(about 60%) without GNN-based self-supervision. We also plot the tSNE graph for the other three datasets in Appendix D. More experiments results with three additional datasets: optdigits, MNIST, and USPS.
>
> To better evaluate the performance improvements, We transform the figure of the ablation studies with GNN-supervision (GFS) and without GNN-supervision into the results in Table 1 with relative performance. The results show that compared with the version without self-supervised graph representation learning, our G-FS has an improvement of 1.89%~17.06%(Average 6.52%). These improvements are significant which are verified by the tSNE results in Fig. 3 and Fig A.2(appendix) for different datasets. Furthermore, the performance is evaluated in a fully supervised setting with different classifiers. The improvement ratio is limited by the capability of the classifier and the problem's complexity. From Table 1 and Table A.2(Appendix), we shows that G-FS achieves good advantages on the two classifiers, LightGBM and Catboost. Therefore, we believe that the improvements are real and convincing.
>
> Table 1: Performance improvement (%) compared with G-FS-g.
>
> | Dataset | Improvement |
> | --- | --- |
> | MNIST(↑) | 3.72% |
> | Optdigits(↑) | 1.89% |
> | USPS(↑) | 2.07% |
> | Concrete(↓) | 12.30% |
> | Protein(↓) | 2.12% |
> | CPU_act(↓) | 17.06% |
> | **Average**| 6.52% |

---

> > ### Comment · Reviewer_s9w1 · 2023-11-22
> > **Response**
> >
> > Thanks for your response. I think my concerns haven't be fully addressed. Considering the efficiency and performance, the improvement of the proposed method is not significant. I will change my rating to 5: marginally below the acceptance threshold.

---

### Official Review · Reviewer_AraV · 2023-10-30

**Soundness:** 2 fair
**Presentation:** 2 fair
**Contribution:** 1 poor
**Rating:** 3
**Confidence:** 4

**Summary:**

This paper study a traditional problem feature selection. It first revisits several remaining issues limit the capability of existing self-supervision enhanced feature selection methods: Then, it proposes a novel method G-FS which performs feature selection based on the discovery and exploitation of the non-Euclidean relations among features and samples by translating unlabeled “plain” tabular data into a bipartite graph. Finally, it conducts some experiments to evaluate the proposed method, showing that the proposed method sometimes outperforms baselines on several tasks across multiple datasets.

**Strengths:**

1.	It tests on several widely-used datasets, and the proposed method can sometimes beat the existing methods.

**Weaknesses:**

1.	The core part (the proposed method in Section 3) lacks of sufficient analysis. We know that it is not difficult to put different modules together to form a paper. But we should make sure that the motivation of doing that really makes sense and we should understand what we are doing.
2.	Confusion of symbol system. For example, the “onehot” in Eq. 5.
3.	The writing needs to be largely improved. The content in introduction is hard to follow. It is also hard to follow the content in Section 3.

**Questions:**

1.	See the weakness in the “*Weaknesses” part.

---

> ### Author Response · Authors · 2023-11-13
> **Response to Reviewer AraV**
>
> Thank you for all your suggestions, we will answer your questions one by one regarding these weaknesses/problems.
>
> **Weakness 1&3:** The core part (the proposed method in Section 3) lacks of sufficient analysis. We know that putting different modules together to form a paper is not difficult. But we should make sure that the motivation of doing that really makes sense and we should understand what we are doing. The writing needs to be largely improved. The content in introduction is hard to follow. It is also hard to follow the content in Section 3.
>
> **Answer to Weakness 1&3:** Sorry for this concern. However, as agreed by Reviewer KtqB and n2md, the motivation of G-FS is to learn more types of relation in the tabular data to support feature selection. According to your suggestion, we have added a paragraph to better illustrate our motivation and reorder the Sect. 3 to make it consistent with Fig. 2. Detailed changes can be found in Reply to Reviewer n2md.
> 1. Thanks for pointing out this, for the introduction, we have modified the Introduction of the paper to make the motivation of the paper more obvious and added more analysis of the proposed method.
> 2. For Section 3, the notation (Section 3.1) and architectural design (Section 3.2) are easy to understand, so we think you are hard to follow about bipartite graph representation learning (Section 3.3). We checked Section 3.3 again and found that the description in this section was inconsistent with the workflow in Figure 2, which made our work difficult to understand. Therefore, we adjusted the order of the chapters and subtitles to make the workflow consistent with Figure 2.
>
> In the revised version:
>
> **3.3 Bipartite Graph Representation Learning**
>
> &nbsp; **3.3.1 Tabular data to bipartite graph**
>
> &nbsp; **3.3.2 GRL for the bipartite graph**
>
> **3.4 Batch-attention-based feature selection**
>
> **Weakness 2:** Confusion of symbol system. For example, the “onehot” in Eq. 5.
>
> **Answer to Weakness 2:** Sorry to make you have this impression. Here, "onehot" is a common method for dealing with categorical data in machine learning, in which a unique binary value represents each distinct category. This is a very popular method. Thus, we use it without a formal definition to avoid page limits.

---

### Official Review · Reviewer_KtqB · 2023-10-30

**Soundness:** 3 good
**Presentation:** 2 fair
**Contribution:** 3 good
**Rating:** 6
**Confidence:** 4

**Summary:**

This article presents an innovative, self-supervised method for augmenting raw features through graph representations. This approach facilitates interaction between samples and features and employs a batch-attention mechanism for feature selection by assigning weights to each dimension of the augmented feature. More specifically, each feature dimension is represented as a unique node, and each sample is also depicted as a unique node with the raw feature of that sample serving as the node feature. Each edge signifies that a sample possesses the edge attribute value at that particular feature dimension. The goal of the self-supervised method is to predict edge attribute, which finally generated augmented features. Experimental results demonstrate that this novel method surpasses traditional benchmarks in most tasks and datasets.

**Strengths:**

1.This paper presents a highly innovative and inspiring approach to representing tabular data and implementing self-supervised learning for augmentations.

2.The proposed method primarily consists of two components: sample-sample interaction and sample-feature interaction.

3.Empirical experiments have been meticulously conducted to compare performance with baseline models and to investigate the effectiveness of the two main components.

**Weaknesses:**

1.Certain equations, like Eq (1) where the sum notation of j is missing, and Eq (4) which doesn't accurately reflect its preceding description, may lead to confusion or mistakes.

2.In Table 1, the arrow for the Testator dataset should be pointing upwards.

3.The paper fails to establish a strong connection between the proposed method and semi-supervised learning, as all referenced methods show a similar performance improvement trend with an increase in labeled data.

4.The section titled "Feature selection under different GNN architecture" doesn't discuss the number of GNN layers, which could potentially lead to oversquashing issues. Furthermore, given that we have two types of nodes (sample and feature), it may not be equitable to compare this with a homophily-based GNN model such as GCN.

**Questions:**

1.How will the different number of the layer of GNN influence the performance? Is there any possible oversquashing or heterophily related problem for your proposed graph building method?

2.What is difference or significance of this method applied in semi-supervised learning and normal supervised learning?

---

> ### Author Response · Authors · 2023-11-13
> **Response to Reviewer KtqB**
>
> **Weakness 1:** Eq (1) where the sum notation of j is missing, and Eq (4) which doesn't accurately reflect its preceding description.
>
> **Answer to Weakness 1:**  For Eq (1), thanks for pointing out this mistake, we are sorry for this and we have corrected it in the revised version.  For Eq (4), we think it is correct. As the masked tabular data might contain both continuous and discrete values for imputation, we use CE loss for discrete attributes (α=1) and MSE loss for continuous attributes (α=0). Thus, we think this formula is consistent with the description.
>
> **Weakness 2:** In Table 1, the arrow for the Testator dataset should be pointing upwards.
>
> **Answer to Weakness 2:** We guess that you are referring to the Tecator dataset, as we did not use the "Testator" dataset. Tecator is a dataset for regression. MAE (lower is better) is used for evaluation. So, the downwards arrow is right.
>
> **Weakness 3:** The paper fails to establish a strong connection between the proposed method and semi-supervised learning, as all referenced methods show a similar performance improvement trend with an increase in labeled data.
>
> **Answer to Weakness 3(Question 2):**
> 1. First, our method belongs to semi-supervised feature selection in the feature selection part. The self-supervised part (without labels) is used to learn the rich latent information in the tabular data. Figure 1 in the Introduction section shows the motivation for our work. The supervision part is used to obtain feature importance scores. We added additional experiments on the Optdigits and USPS datasets, and the results show that G-FS achieves similar feature selection performance with 1/10 the number of labels. Therefore, we believe that G-FS belongs to semi-supervised feature selection, which reduces the label dependence of supervised feature selection through self-supervision.
> 2. However, for the evaluation parts, the quality of selected features is evaluated with fully-supervised settings. Here, LightGBM is adopted. Its performance is heavily dependent on the number of labels. Thus, the three compared methods show an increasing trend. However, with the same number of labels for feature selection, G-FS achieves the best performance at different levels of labeled numbers thanks to its rich relations discovery capabilities of graphs, as explained in Section 4.1.1, 'Why does G-FS work'.
>
> **Weakness 4:** The section titled ... doesn't discuss the number of GNN layers? ...it may not be equitable to compare this with a homophily-based GNN model such as GCN.
>
> **Answer to Weakness 4(Question 1):**
> 1. Thanks for your questions. We have added the discussion about feature selection performance under different GNN layers, please refer to Table 1. The results show that the performance reaches the optimal level on the 3-layer GNN. This is because the 3-layer contains all three types of relations (sample-feature-sample-feature ) , much more than the 1-layer can express (sample-feature) relationship. GNN with 4/5 layers is too complex and may cause the GNN over-smoothing problem.
> 2. Then, you proposed an important question 'This comparison is not equitable with a homophily-based GNN model such as GCN'. Yes, this does seem unfair, but what we want to illustrate in this experiment is that our method pays more attention to the balance of integrity and consistency than GCN, which is crucial for feature selection. GCN is more inclined to data consistency, while GraphSAGE combines sampling and aggregation to learn richer representations in graphs. So we can see that the result of G2SAT is better.
> 3. As far as we know, there are few self-supervised learning methods based on heterogeneous bipartite graphs. Here we compared with a recently heterophily-based GNN model IGRM[1], which is more considered to the diversity of nodes. The result in Table 2 shows that too much bias towards homogeneity and heterophily will lead to a decrease in the performance of feature selection. How to find a suitable balance between homogeneity and heterogeneity will be a very interesting research content.
>
> Table 1: Performance comparison under different GNN layers.
>
> | GNN_layer | Optdigits(↑) | USPS(↑) | Concrete(↓) | Tecator(↓) |
> | --- | --- | --- | --- | --- |
> | 1 | 73.46±3.45 | 80.89±2.74 | 6.18±0.52 | 1.16±0.19 |
> | 2 | 76.82±3.01 | 81.65±1.82 | 5.31±0.40 | 1.08±0.11 |
> | 3(Ours) | **79.08±2.57** | **83.10±1.46** | **4.96±0.29** | **1.03±0.17** |
> | 4 | 75.67±2.72 | 81.20±2.61 | 5.46±1.26 | 1.06±0.14 |
> | 5 | 69.90±4.13 | 79.60±2.47 | 7.34±2.53 | 1.11±0.13 |
>
> Table 2: Performance comparison under different GNN structures.
>
> | Structure | Optdigits(↑) | USPS(↑) | MBGM(↓) | Tecator(↓) |
> | --- | --- | --- | --- | --- |
> | G-FS(G2SAT) | **79.08±2.57** | **83.10±1.46** | **5.45±0.11** | **1.03±0.17** |
> | G-FS(EGCN) | 72.50±3.56 | 81.14±2.07 | 5.90±0.35 | 1.16±0.14 |
> | G-FS(IGRM) | 75.51±2.02 | 80.71±0.98 | 6.12±0.44 | 1.08±0.21 |
>
> [1] Data imputation with iterative graph reconstruction, AAAI 2023

---

### Official Review · Reviewer_n2md · 2023-10-31

**Soundness:** 3 good
**Presentation:** 2 fair
**Contribution:** 2 fair
**Rating:** 5
**Confidence:** 3

**Summary:**

The paper studies the problem of semi-supervised feature selection. The authors introduce a graph representation learning-enhanced semi-supervised feature selection framework called G-FS, which transforms unlabeled tabular data into a bipartite graph to discover relationships between features and samples. They design a self-supervised edge prediction task to convert rich information on the graph into low-dimensional embeddings, reducing redundant features and noise. Additionally, the authors propose a batch-attention feature weight generation mechanism, which can generate more robust weights.

**Strengths:**

- The idea of enhancing semi-supervised feature selection using graph representation learning is novel.
- Mapping tabular data to a bipartite graph allows for the exploration of more potential relationships.

**Weaknesses:**

- The writing of the paper can be further improved.
- The improvements over existing methods are not significant.

**Questions:**

- In Figure 1b, the authors depict the sample-label relationship. However, this relationship is not mentioned in the text. In the introduction, the authors mention three types of relationships, yet in Figure 1c, four types of relationships are referenced.
- Minor errors: In Section 4.2, the reference to Table 1 is incorrectly written as Figure 1. (To verify the performance of G-FS, G-FS is compared with other feature selection methods on 12 different datasets (refer to Figure 1).)

---

> ### Author Response · Authors · 2023-11-13
> **Response to Reviewer n2md**
>
> Thank you for all your suggestions, we will answer your questions one by one regarding these weaknesses/problems.
>
> **Weakness 1:** The writing of the paper can be further improved.
>
> **Answer to Weakness 1:** Thanks for your pointing out this question. We have revised the Introduction and Methods Sections of the manuscript to make the purpose of the paper more evident and the proposed method easier to comprehend.
>
> Specifically, **1)** we explicitly state our design motivation in a newly introduced subsection, 'motivation', with Figure 1 showing the four types of relations in the tabular data. Thus, we propose using graphs and GNN to represent and learn those relations, as autoencoder-based self-supervision can hardly express and learn those relations. **2)** we adjusted the order of the subsections of Sect. 3.3. By putting the translation process from the tabular data to the bipartite graph upfront, the order of Section 3.3 is consistent with Figure 2. We hope it is easier to understand.
>
> **Weakness 2:** The improvements over existing methods are not significant.
>
> **Answer to Weakness 2:** Thanks for your questions, we will answer your questions in two parts, overall performance and performance under different scenarios.
> 1. For typical supervised classification and regression tasks, G-FS achieves performance improvements.  For classification tasks, we achieved the best performance on different types of datasets with an average of 20.32% than all baselines and led the second strong baseline 1.13%\~19.50%.  We guess your concerns are about the results of the regression tasks. The regression task is relatively simple and most FS baselines achieve similar results. However, G-FS still achieves good results, especially for MBGM and Pdgfr (two difficult datasets with dim>300), G-FS leads 2.93%\~9.92%. For CPU, Protein, and Concrete, they are comparably simple. G-FS still leads 0.82%~4.22% in MAE than the second best XGBoost. XGBoost is a very strong baseline in the regression task [1]. Many recent FS do not compare the regression datasets with XGBoost. If XGBoost is excluded, our performance gain can be further improved.
> 2. For other scenarios, such as one-shot, noise disturbance, synthetic data, and limited labeled/unlabeled samples,  G-FS achieves more significant leads compared with other methods. Thus, we think G-FS can be applied to different scenarios and its performance improvements are consistent, albeit with different degrees in different problems.
>
> [1] Shwartz-Ziv R, Armon A. Tabular data: Deep learning is not all you need[J]. Information Fusion, 2022, 81: 84-90.
>
> **Question 1:** In Figure 1b, the authors depict the sample-label relationship. However, this relationship is not mentioned in the text. In the introduction, the authors mention three types of relationships, yet in Figure 1c, four types of relationships are referenced.
>
> **Answer to Question 1:** For the feature-label relations, we did mention this relation(maybe implicit) in the text in the introduction of supervised feature selection 'in discriminative information encoded in class labels or regression targets', we apologize for the ambiguity. We have updated our manuscript to mention it explicitly in the introduction.
>
> **Question 2:** Minor errors: In Section 4.2, the reference to Table 1 is incorrectly written as Figure 1. (To verify the performance of G-FS, G-FS is compared with other feature selection methods on 12 different datasets (refer to Figure 1).)
>
> **Answer to Question 2:** Thanks for pointing out these errors; we are sorry for these errors. The description of 'four types' in Figure 1c is right, we have corrected it and other minor errors in the revised manuscript.

---

### Author Response · Authors · 2023-11-21

Dear Reviewers:

We write to remind you about reviewing the rebuttal for the paper. We sincerely appreciate the time and effort you have already invested in reviewing the initial submission. We prepared a comprehensive rebuttal and a revised manuscript response addressing all the comments. Your expertise and insights are highly valuable in evaluating our responses. As the Author-Reviewer discussion ends on November 23, we are eager to receive your feedback. If you have any questions or require any further information, please feel free to reach out to us.

Kinds Regards,

Authors

---

### Meta-Review · Area_Chair_WDiQ · 2023-12-06

**Metareview:**

In this paper, the authors proposed Graph representation learning enhanced Semi-supervised Feature Selection (G-FS) for tabular datasets. G-FS combines a self-supervised edge prediction module to smooth the original tabular dataset and a batch-based attention generation module by (Liao et al., 2021) to identify important features under semi-supervised settings.

 I recommend to reject this paper due to the weakness pointed by reviewers: 1) the novelty of G-FS is limited & lack of sufficient analysis (Reviewer AraV), 2) presentation/writing of the paper requires improvement (Reviewer n2md, Reviewer AraV), and 3) The improvements over existing methods are not significant given the trade-off of efficiency (Reviewer n2md & Reviewer Re  viewer s9w1).

**Justification For Why Not Higher Score:**

I have read the paper and agreed with the majority of reviewers that the paper is lack of enough novelty (it is essentially a combination of two known techniques).

**Justification For Why Not Lower Score:**

N/A

---

### Decision · Program_Chairs · 2024-01-16

Reject